



# Annual and interannual variability and trends of albedo for Icelandic glaciers.

Andri Gunnarsson[1,4], Sigurdur M. Gardarsson[1], Finnur Pálsson[2], Tómas Jóhannesson[3], and Óli G.B. Sveinsson[4]

[1]University of Iceland, Civil and Environmental Engineering, Hjardarhagi 2-6, IS-107 Reykjavik, Iceland
[2]Institute of Earth Sciences, University of Iceland, Sturlugata 7, 101 Reykjavík, Iceland
[3]Icelandic Meteorological Office, Bústaðavegi 7–9, 180, Reykjavík, Iceland
[4]Landsvirkjun, Department of Research and Development, Reykjavík, IS-107, Iceland

**Correspondence:** Andri Gunnarsson (andrigun@lv.is)

**Abstract.**

During the melt season, absorbed solar energy, modulated at the surface predominantly by albedo, is the governing factor controlling surface-melt variability for glaciers in Iceland. Using MODIS satellite-derived daily surface albedo, a gap-filled temporally continuous albedo product is derived for the melt season (MJJA) for the period 2000–2019. The albedo data are thoroughly validated against available in-situ observations from 20 glacier automatic weather stations for the period 2000–2018. The results show that spatio-temporal patterns for the melt season have generally high annual and inter-annual variability for Icelandic glaciers, ranging from high fresh- snow albedo of about 85–90% in spring, decreasing to 5–10% in the impurity-rich bare-ice area during peak melt season. The analysis shows that the volcanic eruptions in 2010 and 2011 had significant impact on albedo and also had a residual effect in the following years. Furthermore, airborne dust, from unstable sandy surfaces close to the glaciers, is shown to enhance radiative forcing and decrease albedo. A significant positive albedo trend is observed for northern Vatnajökull while other glaciers have non-significant trends for the study period. The results indicate that the high variability in albedo for Icelandic glaciers is driven by climatology, i.e. snow metamorphosis; tephra fall-out during volcanic eruptions and their residual effects in the post-eruption years; and dust loading from widespread unstable sandy surfaces outside the glaciers. This illustrates the challenges in albedo parametrization for glacier surface-melt modelling for Icelandic glaciers as albedo development is driven by various complex phenomena, which may not be correctly captured in conventional energy-balance models.

## 1 Introduction

Surface albedo is defined as the unitless ratio of radiant flux reflected from the Earth's surface to the incident flux. It is a controlling parameter, which governs the portioning of the shortwave radiative energy between the atmosphere and the surface and,





therefore, a control of the surface energy balance modulated by the solar zenith angle, cloud optical thickness, cloud cover and transmission properties of the atmosphere (Gardner and Sharp, 2010; Klein and Stroeve, 2002; Donohoe and Battisti, 2011). The evolution of albedo for impurity-free snow and ice is controlled by the snow metamorphism process where snow-grain size increases with time and lowers albedo at all wavelengths while fresh new snow increases albedo (Warren, 1982). Light
Absorbing Particles (LAP) such as sand, mineral and volcanic dust/tephra, black carbon and algae in the near-surface layers of the snow and ice further lower the albedo enhancing the energy absorbed by the surface (Warren and Wiscombe, 1980; Box et al., 2012; Painter et al., 2012; Stibal et al., 2017; Skiles et al., 2018).

Iceland is located in the North-Atlantic Ocean on the mid-Atlantic Ridge, close to the Arctic Circle (between 63° and 66°
N) with an area of 103.100 km$^2$. At present, about 10% (10.344 km$^2$) of Iceland is glaciated (Fig. 1) (Björnsson and Pálsson, 2008). Icelandic glaciers span an elevation range from sea level up to 2110 m a.s.l. in an maritime climate, with large mass throughput and high variability in annual mass balance (Björnsson and Pálsson, 2008; Þorsteinsson et al., 2017; Pálsson et al., 2019, 2020).

Glacier research is important in Iceland for multiple reasons. Seasonal glacier melt is vital for hydropower production and
melt water storage in reservoirs as the energy system is strongly dependent on glacier and snow melt providing over 72% of the total average energy produced in Iceland (Hjaltason et al., 2018). The system isolation and high natural climate variability can pose a risk to the reliability of the energy system as drought conditions, low-flow periods and years with low summer mass balance are challenging to foresee. Activity in glacier-covered volcanoes can cause volcanic ash and tephra fall-outs on glaciers during explosive eruptions leading to enhanced melt or in some cases glacier surface isolation reducing
melt significantly (Möller et al., 2014; Wittmann et al., 2017a; Möller et al., 2019). For Icelandic glaciers, surface albedo are the dominating factors governing surface melt annual variability (De Ruyter De Wildt et al., 2002; Guðmundsson et al., 2009) and the importance of correct representation of surface albedo for glacier melt modelling is critical (Schmidt et al., 2017).

Optical satellite remote sensing offers a way to observe surface albedo continuously at large spatio-temporal scales but are
limited to times of clear-sky overpasses. Various studies have shown that surface albedo over snow and ice can be successfully derived from visible and near-infra red satellite sensors (Stroeve et al., 1997; Reijmer et al., 1999; Stroeve, 2001; Klein and Stroeve, 2002; Liang et al., 2005; Stroeve et al., 2005, 2013). Since October 1978, regular polar coverage has been provided by the National Oceanographic and Atmospheric Administration (NOAA) satellites carrying the advanced very high-resolution (AVHHR) radiometers (Stroeve et al., 1997; Xiong et al., 2018). The AVHHR sensor has visible, near-infrared and thermal
channels that observes the top of the atmosphere (TOA) radiances under clear-sky conditions which allows for conversions of narrow-band reflectance measurements to broadband albedo by applying an atmospheric correction and using an radiative transfer model with successful results over snow- and ice-covered surfaces (Lindsay and Rothrock, 1994; de Abreu et al., 1994; Stroeve et al., 1997; Reijmer et al., 1999). Spatial resolution is 4 and 1.1 km depending of the collection mode (global or local) allowing for sufficient representation of surface albedo for larger ice caps or sheets that encompass large areas such as
Greenland (Steffen et al., 1993; Zhou et al., 2019) and the main ice caps of Iceland. Higher spatial-resolution optical data have



been obtained from the Landsat constellation (30 m spatial resolution) for albedo retrievals with capabilities to further resolve smaller-scale patterns, more detailed variability of albedo and sub-pixel variability of large-footprint satellite sensors (Winther, 1993; Reijmer et al., 1999; Gascoin et al., 2017; Naegeli et al., 2017, 2019) . Higher spatial-resolution satellite data generally have the disadvantage of lower temporal resolution, which excludes the possibility of daily albedo observations.


Since February 2000, the Moderate Resolution Imaging Spectroradiometer (MODIS) instrument, on board the NASA TERRA satellite, has collected daily multi-spectral radiance data (36 spectral bands) viewing the entire Earth's surface every 1 to 2 days at a 500 m spatial resolution. Followed by the NASA AQUA satellite launch in July 2002, carrying the MODIS sensors as well, MODIS data have significantly improved understanding of global-earth and lower-atmosphere processes and

dynamics. Various albedo products for snow- and ice-covered surfaces have been developed and analysed to further understand the inter-annual and seasonal variability in albedo for glaciers and ice sheets (Stroeve et al., 2005; Box et al., 2012; Möller et al., 2014; Gascoin et al., 2017).

Reijmer et al. (1999) found that the temporal and spatial variations in the surface albedo of the Vatnajökull ice cap was

reproduced fairly well by using AVHRR data for the melt season in 1996. To confirm this hypothesis, in-situ data and higher resolution data from Landsat 5 Thematic Mapper (TM) sensor were compared as well showing greater variability in surface albedo implying that the scale of the albedo variations is larger than the AVHHR pixel (1.1 km) could resolve. De Ruyter De Wildt et al. (2002) assessed Vatnajökull glacier albedo using AVHRR images and found a strong correlation ($R^2$: 0.87–0.94) between the mean albedo of the entire ice cap through the melting season to observed specific mass balance for the period

from 1991–99. In the accumulation area, average albedo was found to decrease from 80% down to 60% with a gradual decrease during the melt season, while in the ablation area, values as low as 10% ranging up to 35% with considerable variation were found. Gascoin et al. (2017) indicate a good ability of the MODIS MCD43A3 multi-look product to characterize seasonal and interannual albedo changes with correlation coefficients ranging from 0.47 to 0.90 but high RMS errors in comparison with in-situ data. Subpixel variability was also investigated using Landsat 5 and 7 data similar to Reijmer et al. (1999) with generally

better results. Möller et al. (2014) investigated the influence of tephra depositions from the 2004 Grímsvötn eruption in Vatnajökull glacier by using the MODIS MCD43A3 multi-look product in combination with daily observations from the MOD10A1 product. By developing an empirically-based model to describe the albedo decrease across the glacier surface caused by the deposited tephra, they concluded that the tephra-induced albedo changes were largest and most widely distributed over the glacier surface during the summer season 2005 when the observed albedo decrease reached 35% as compared with modelled

undisturbed conditions. A study by Wittmann et al. (2017a) for the 2012 melt season, states that the positive radiative forcing of airborne dust on Brúarjökull can add up to an additional 1.1 m w.e. (water equivalent) of snowmelt (42%) compared with a hypothetical clean glacier surface. This represents the influence of volcanic eruptions and airborne dust deposits on the mass balance of Icelandic glaciers. In most cases, dust and tephra will amplify surface melt due to additional radiative forcing during the melt season but in some cases, ash layers exceeding a certain critical thickness can cause insulation of the underlying snow

and ice. Results by Dragosics et al. (2016) found this critical thickness to range from 9–15 mm dependant on grain size and

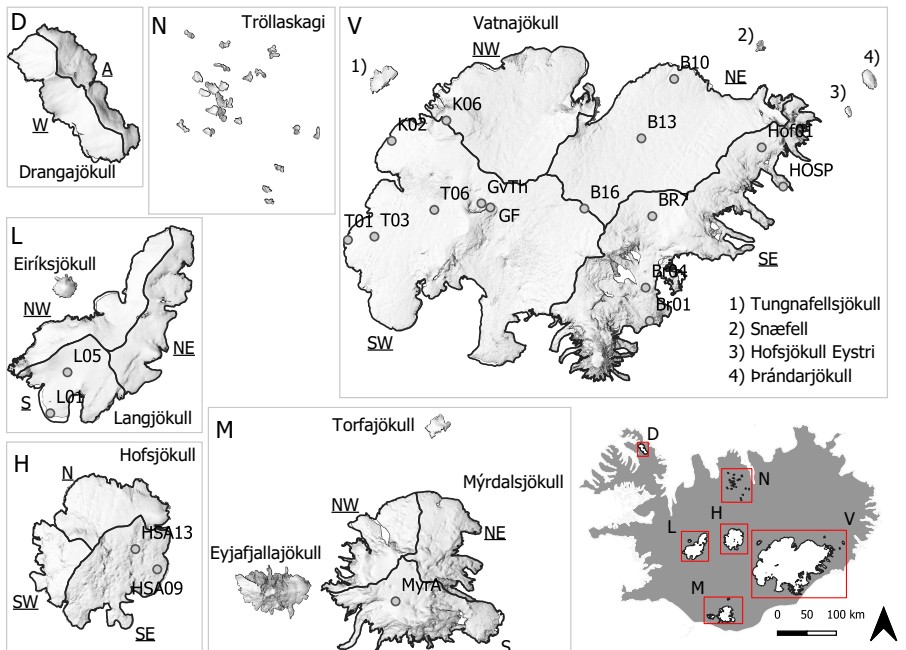

**Figure 1.** Location map of Icelandic glaciers used in the study. These are glaciers that were at least $2 \, \text{km}^2$ or eight unmixed MODIS pixels. For the larger glaciers, Vatnajökull, Langjökull, Hofsjökull, Mýrdalsjökull and Drangajökull, smaller areas are defined to the main ice flow basins of the glaciers for further analysis. These delineated areas are annotated with underlined text (e.g. NW for northwest). In total, 28 areas are processed, including the sub-areas, but small mountain glaciers in northern Iceland were merged into one processing unit. Available glacier automated weather stations are shown with grey dots. Further details of these stations are given in Table 1. A shaded relief representation of a glacier DEM is from Jóhannesson et al. (2013) and catchment delineation from Magnússon et al. (2016), for Drangajökull, Björnsson (1988) and (Björnsson et al., 2000) for Hofsjökull and Mýrdalsjökull, and Pálsson et al. (2013, 2016) for Langjökull and Vatnajökull.

material type.

The aim of this study was to create a gap-filled MODIS-based surface-albedo product for glaciers in Iceland for the time period from 2000 to 2019, validated with in-situ data. The resulting gap-filled product was then used to analyse and quantify, spatio-temporal patterns of albedo for Icelandic glaciers for the time period.

## 2 Data and Methods

Figure 1 shows the location map of Icelandic glaciers used in the study. These are glaciers that were at least $2 \, \text{km}^2$ or eight unmixed MODIS pixels. For the larger glaciers, Vatnajökull, Langjökull, Hofsjökull, Mýrdalsjökull and Drangajökull, smaller areas were defined to represent the main ice flow basins of the glaciers for more detailed analysis.



## 2.1 MODIS products

Daily snow cover data products calculated from the MODIS spectroradiometer on the NASA TERRA (MOD10A1 V006) and AQUA (MYD10A1 V006) platforms were obtained from the National Snow and Ice Data Center (NSIDC). The products provide daily estimates of snow cover, blue-sky albedo and a quality assessment at 500 m spatial resolution for cloud free conditions at the satellite platform overpass (Hall and Riggs, 2016a, b). Daily albedo calculations use reflectances of the first seven visible and near-infrared bands of the MODIS spectroradiometer (459–2155 $\mu$m) which have been corrected for atmospheric effects. To correct for anisotropic scattering effects of snow and ice the DIScrete Ordinates Radiative Transfer model (DISORT) is applied. The daily estimated blue-sky albedo corresponds to the broadband albedo for actual direct and diffusive illumination. (Klein and Stroeve, 2002), and is therefore, directly comparable to field observations with broadband radiometers (Stroeve et al., 2013). For comparison and validation purposes, the multi-look MCD43A3 albedo product V006 was obtained as well from LP DAAC (Schaaf and Wang, 2015). MCD43A3 provides daily albedo using 16 days of Terra and Aqua MODIS data at 500 meter (m) resolution. Data are temporally weighted to the ninth day of the 16 day. The MCD43A3 product provides black-sky albedo (directional hemispherical reflectance) and white-sky albedo (bihemispherical reflectance) data at local solar noon for the same bands as used in M*D10A1 albedo products.

The quality of remotely-sensed albedo retrievals decreases during fall and winter as the incoming solar irradiance and solar incidence angle decreases. With an increase in solar zenith angles (SZA) and especially beyond 70° the accuracy of satellite- and ground-based instruments declines for albedo retrievals. This results in cases where unrealistic and unexpected values are observed and often exceed expected maximum clear-sky snow albedo. Due to polar darkness (SZA >85°), MODIS data are generally not available from mid-November until mid-January each year over Iceland (Dietz et al., 2012). Cloud cover in Iceland also poses a challenge in using optical remote sensing as average cloud cover ranges from 70–90% with little inter-annual variability (Gunnarsson et al., 2019).

The scope of this study was limited to the melt season of Icelandic glaciers when SZA are low and incoming solar irradiance high (May, June, July and August). Every granule from MODIS tile h17v02 was used in this project as it covered all the central highlands in Iceland and left out only a small portion of the western Snæfellsnes peninsula and the Westfjords.

## 2.2 Meteorological in-situ data

The Icelandic Glacier Automatic Weather Stations network (ICE-GAWS) provided automatic weather-station observations from Vatnajökull, Langjökull, Hofsjökull and Mýrdalsjökull since 1994, 2001, 2016 and 2015, respectively. Most stations in the network were operated during the extended melt season (MJJASO) annually while a few sites were operated all year around. All sensors were tested and validated annually before deployment in the field in spring. Location of the sites are shown in Figure 1 and location, elevation, observation period and radiometer instrumentation in Table 1.



Radiation was measured with a net radiometer equipped with two pyranometers facing upward and downward, respectively, used to measure the incident (SW↓) and reflected shortwave radiation (SW↑) as an 10 minute average. The ratio of both quantities allowed the bi-hemisperical albedo of the surface to be estimated. For comparison purposes in this study, daily integrated

albedo is used instead of selecting the hourly-mean albedo measured closest in time to the satellite overpasses. Daily integrated albedo was calculated as the running 24-hour sum of upward shortwave divided by the running 24-hour sum of the downward shortwave. This method minimizes the effect of solar zenith angle on the accuracy of the albedo estimation and is less sensitive to radiometer level and cosine response errors since it integrates errors that partly cancel each other (Box et al., 2012). Daily integrated albedo has been shown to represent the daily variability of the glacier surface but only partially represent diurnal

variability, such as onset of melt (Stroeve et al., 2005).

Most sites in the GAWS network used Kipp and Zone CM14, CNR1 and CNR4 radiation sensors which have relatively uniform spectral response ranging from 0.3-–2.8 $\mu m$ with uncertainty that has been reported to be 3—10% for daily totals over ice- and snow-covered surfaces (Van Den Broeke et al., 2004; Guðmundsson et al., 2009; Kipp and Zonen, 2019). The LI-COR

200 SZ pyranometers were used at a few sites. They have reduced spectral response (0.4—1.1 $\mu m$) compared with the Kipp and Zone instruments. Tilting of the instruments with respect the glacier surface was not monitored and could add further to the uncertainty especially in the ablation zone of the glaciers (Van den Broeke et al., 2004). The incoming and reflected shortwave measurements from 20 AWSs during the period 2000–2018 were used to validate the MODIS remotely-sensed albedo products.

Manual quality control of the data was done by screening shortwave and albedo data and remove obvious errors, periods when stations are buried in snow and calibration periods prior to site installment in spring. Obvious cases of instrument failure were also rejected. Observations of upward solar irradiance exceeding downward solar irradiance were also removed. Quality control was carried out on the data at an hourly time step prior to aggregating to daily and monthly time steps. Daily averages were calculated from hourly averages if at least 20 hourly values were available and monthly averages were calculated from

daily averages if 24 values or more were available.

### 2.3  Data processing

### 2.4  MODIS data processing

From the MOD10A1 and MYD10A1 daily data tiles, the *MOD Grid Snow 500 m* grid and the grid variable *Snow Albedo Daily Tile* were used for the albedo analysis. Snow albedo is reported in the range 0–100 where the snow/ice-cover mask in the

M*D10A1 product identifies whether a pixel is snow covered or not. A processing pipeline for MODIS snow-albedo data was partly adopted from Box et al. (2012) with modifications and adoptions for Icelandic glaciers.

Temporal aggregation was applied to the MOD10A1 and MYD10A1 data to reduce the number of unclassified daily pixels due to clouds at the overpass time. The temporal aggregation range was set as the number of days backwards and forwards at each center date (t = 0) to merge to a single stack for further processing. A temporal aggregation range as 5 days backward/forward



**Table 1.** Overview of average location, elevation, average operating period and radiometer instrument of the GAWS network used for validation. All stations have temperature probes while GV (Grímsvötn) and GF (Grímsfjall) only observe temperature and not short wave irradiation. Location and elevation is based on the average location of the site for the observation period.

| Site | Glacier outlet | Latitude | Longitude | m a.s.l. | Operation | Radiometer |
|------|----------------|----------|-----------|----------|-----------|------------|
| Kokv | Vatnajökull SW | 64.589 | -17.860 | 1096 | MJJAS | LiC |
| BRE | Vatnajökull SE | 64.094 | -16.325 | 210 | MJJAS | CNR1 |
| B10 | Vatnajökull NE | 64.728 | -16.112 | 779 | All year | CNR1/CNR4 |
| B13 | Vatnajökull NE | 64.576 | -16.328 | 1216 | MJJASO | CM14/CNR4 |
| B16 | Vatnajökull NE | 64.402 | -16.681 | 1526 | MJJASO | CNR1 |
| BRE1 | Vatnajökull SE | 64.097 | -16.329 | 116 | All year | CNR1 |
| BRE4 | Vatnajökull SE | 64.183 | -16.335 | 529 | MJJASO | CNR1 |
| BRE7 | Vatnajökull SE | 64.369 | -16.282 | 1243 | MJJASO | CNR1 |
| T01 | Vatnajökull SW | 64.326 | -18.118 | 772 | All year | CNR4 |
| T03 | Vatnajökull SW | 64.337 | -17.977 | 1069 | MJJASO | CNR1 |
| T06 | Vatnajökull SW | 64.404 | -17.609 | 1466 | MJJASO | CNR1 |
| K06 | Vatnajökull SW | 64.639 | -17.523 | 1946 | MJJASO | CM14 |
| MYRA | Mýrdalsjökull | 63.612 | -19.158 | 1346 | MJJAS | CM14 |
| HSA09 | Hofsjökull SE | 64.770 | -18.543 | 840 | MJJASO | CNR1 |
| HSA13 | Hofsjökull SE | 64.814 | -18.648 | 1235 | MJJASO | CNR1 |
| L05 | Langjökull S | 64.595 | -20.375 | 1103 | MJJASO | CNR1 |
| SKE02 | Vatnajökull SW | 64.303 | -17.153 | 1208 | MJJASO | CNR1 |
| L01 | Langjökull S | 64.514 | -20.450 | 589 | All year* | CNR1 |
| Hof01 | Vatnajökull SE | 64.539 | -15.597 | 1142 | All year | LiC |
| Hosp | Vatnajökull SE | 64.431 | -15.478 | 76 | MJJASO | LiC |

(t = ±5 d) was selected; i.e., in total 11 days can contribute data to the temporally aggregated product. A total of 22 values are potentially available for each pixel (i.e. 11 days of MOD10A1 and 11 days of MYD10A1). This reduces the number of pixels classified as no data (cloud cover, detector saturation, etc.) by 66%. From the 22 potentially available values, the mean is calculated to represent the surface-albedo, after median based statistical rejection of outliers. Extremly high MODIS albedo values from the original products (MOD10A1 and MYD10A1) ($\alpha$>90) are excluded as these are considered unrealistic values

under clear skies (Konzelmann and Ohmura, 1995; Box et al., 2012) .

Cloud cover is known to be a major challenge in optical remote sensing of the Earth surface, especially for snow- and ice-covered surfaces (Davaze et al., 2018; Gunnarsson et al., 2019). Various methods exist to differentiate between clouds and snow- and ice covered surfaces (Ackerman et al., 1998; Sirguey, 2009) but omission errors are challenging to avoid

completely leading to misclassification of surface albedo and clouds. Manual inspection of the raw MODIS albedo data for





Icelandic glaciers revealed misclassified pixels due to various artefacts such as clouds boundaries, cloud shadows, contrails, cirrus clouds and fog, especially in the glacier terminus area. These artefacts create abrupt changes in the surface-albedo time series making it possible to reject them based on the temporally aggregated data statistics. On a pixel-by-pixel basis the method by Box et al. (2012) was applied to reject values that exceed 2 standard deviations from the 11 day temporally aggregated data

stack. The method is only applied if 4 or more pixels in the data stack have valid albedo data. To prevent rejection of valid data, values that were within a certain threshold from the median were not rejected. The outlier thresholds were manually adjusted mostly related to the elevation of the glaciers, ranging from 1 - 4%, for higher to lower elevation, respectively.

Finally, after temporal aggregation, outlier removal and statistical filtering, the still remaining unclassified pixels were clas-

sified statistically with four predicting variables, location (easting, northing), elevation (Z) and aspect with a daily trained random forrest model (Matlab, 2017). Topographic and masking data for ice-covered surfaces were obtained from the National Land Survey of Iceland. The original digital elevation model was a raster with a 10 m spatial resolution which is resampled to match the grid of the MODIS pixels using bilinear sampling (GDAL/OGR contributors, 2019). Correspondingly for each pixel aspect was calculated. To evaluate the model classification performance 25% of the classified data from the temporal

aggregation were withheld for comparisons purposes. The average RMS error of the classified data was 3.49 with an standard deviation of 0.80 for the period from May to August. On a monthly, basis the lowest RMS error was observed in May ($\mu : 3.17$, $\sigma : 0.80$) and the highest in August ($\mu : 4.03$, $\sigma : 0.83$) while June and July fall in between. For individual years the RMS errors were the highest in 2010 $\mu : 4.02$, $\sigma : 1.42$) and 2011 $\mu : 4.73$, $\sigma : 1.32$) for MJJA averages. This was most likely due to the volcanic eruptions in Eyjafjallajökull in 2010 and Grímsvötn in 2011. This resulted in volcanic tephra depositions on Icelandic

glaciers that poorly correlate to topographic patterns of albedo as the random forest model was trained on location, elevation and aspect. The final output, a daily gap-filled albedo grid, which was used for further processing, is hereafter refereed to as MCD11.

For MCD43A3 multi-look data to be comparable with GAWS data, the blue sky albedo was calculated as the average between

the black-sky albedo and the white-sky albedo tiles in the product, assuming a constant fraction of diffuse illumination as done by Möller et al. (2014) and Gascoin et al. (2017) in previous studies at Icelandic glaciers. For cloud cover estimations, daily valid pixels in MOD10A1 (AM overpass) and MYD10A1 (PM overpass) were merged to a single daily product, representing average daily cloud cover.

To quantify the changes in albedo over time, trends were calculated. The calculations are pixel-based from annual averages

for the period 2000–2019. Significance of data were calculated using the Mann–Kendall test. The Mann–Kendall test is a nonparametric test, that detects the presence of a monotonic tendency in chronological data, and identifies trends in data over time without an assumption of normality (Helsel and Hirsch, 2002). Trends are considered statistically significance then the p-value is lower than 0.05. For this study, glacier boundaries delineated in 2010 and 2012 were used for Vatnajökull, 2007 and 2008 for Langjökull and Hofsjökull, respectively. This was selected as a midpoint representing an average glacier area during the

period 2000–2019. This needs to be considered when discussing rapid changes at the glacier terminus, as some areas in 2000




where part of an active glacier but might in 2019 be dead ice or land.

## 3 Results and discussion

### 3.1 MODIS albedo validation

The MODIS albedo data was validated by a pixel-based comparison, i.e. the nearest pixel to the GAWS station locations
was extracted to a time series. In total 20 GAWS sites have SW↓ and SW↑, enabling albedo calculations during the period
2000–2019 spanning elevations from 100 to 1850 m a.s.l., ensuring validation data over a wide elevation range at Vatnajökull,
Hofsjökull, Langjökull and Mýrdalsjökull. Months with less than 26 days of GAWS data were excluded from the comparison
and daily averages were not calculated unless 22 hours of data within a day were available. MODIS provides only albedo
estimation for clear sky conditions.

Figure 2 shows the comparison results for May, June, July and August for MCD11. Overall good visual and statistical agree-
ment is found between the MODIS MCD11 data and the in-situ albedo from GAWS observations. For the whole period from
May–August, the RMS error is 7.2 with an $R^2$ of 0.9. The GAWS observation network captures a wide range of melt-season
variability of albedo ranging from 6–90% which is well captured with the MODIS MCD11 product as demonstrated with
the overall high correlation coefficients. Based on linear regression (red lines in Figure 2) for all months, albedo was slightly
underestimated for higher values (albedo > ~55) and slightly overestimated at lower values by the MODIS MCD11 product.
Various reasons could contribute to these differences, such as sensor accuracy and instrument installation configuration (i.e.
tilting, riming on the sensor dome). In the ablation zone, where the lowest albedo values were observed, high melt rates (surface
lowering of 3–7 m) can contribute to progressing tilting of the instruments over the ablation period. Large sand and tephra-
covered areas have been observed in the instrument footprint during field visits, as well as melt channels and small melt water
ponds offsetting the spectral properties of the surface compared with the spectral response of snow and ice, inducing errors in
the comparison between in-situ and remotely-sensed albedo. The temporal aggregation of the remotely-sensed data includes a
dampening effect on the MCD11 data compared to the GAWS observations, which could possibly partially explain outliers in
July and August when the in-situ observations are higher than the MCD11. Extensive snowfall events, occurring under cloud
cover and limiting accurate data retrievals by the satellites, will lead to albedo that is not correctly represented in the MCD11
reconstruction due to the 11 day temporal aggregation.

Table 2 shows a comparison of MCD11 with other albedo products from MODIS, i.e. MOD10A1, MYD10A1 and MCD43A3.
In most cases, the MCD11 product had lower RMS errors and higher correlation coefficients indicating the success in removing
spurious values such as misclassified clouds, image stripes and other artifacts in the original MODIS products. No correlation
was found between RMS error and GAWS location (elevation or glacier/location). No adjustments or calibrations are applied
to the MCD11 product in the further use in this study. Table B1 shows validation results for individual stations for MOD10A1,

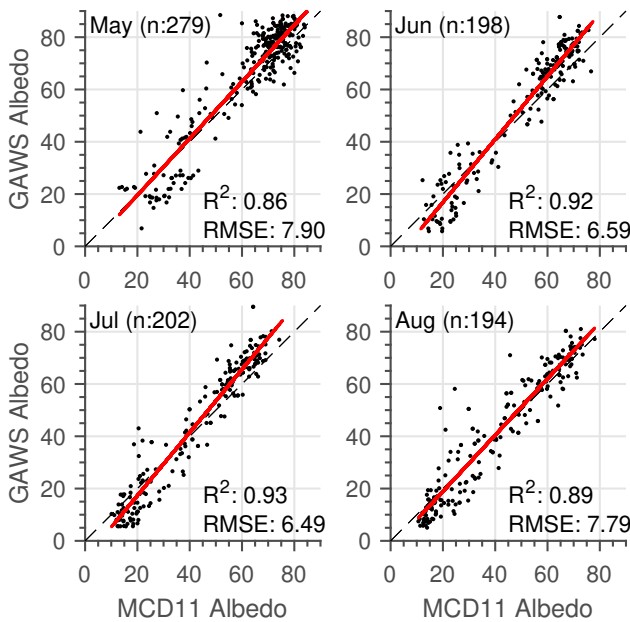

**Figure 2.** Comparison of monthly averaged MODIS albedo with in-situ GAWS albedo observations for May, June, July and August for the period from 2000–2019 where data were available for the MCD11 data product.

MYD10A1, MCD43A3 and MCD11.


The comparison presented here is in fact similar to previous work on Icelandic glaciers by Gascoin et al. (2017) where the MCD43A3 was evaluated with RMS errors ranging from 8–21%. Various studies in Greenland using in-situ AWS report lower RMS errors, ranging from 2.8–5.4% on a monthly basis for MOD10A1 using 17 stations for validation by Box et al. (2012) and a total RMSE of 6.7% in a study by Stroeve et al. (2013) using MCD43A3 high quality retrievals. It is important to consider
how representative point-based in-situ observations are (observing ∼120–180 m$^2$ (Kipp and Zonen, 2019)), compared with the spatial footprint of the MODIS data (0.25 km$^2$), especially in glaciated areas with high spatial albedo variability and MODIS sub-pixel variability as is observed in the bare-ice areas of the Icelandic glaciers. Sub-pixel variability has been investigated by Reijmer et al. (1999) and Gascoin et al. (2017) for the Icelandic glaciers indicating high sub-pixel albedo variability.

**3.2 Gap-filled albedo**

Figure 3 shows the average cloud cover for the main Icelandic glaciers from April to October, based on daily MODIS data from AQUA and TERRA. This highlights the challenges for optical satellite remote sensing in Iceland due to cloud obscurity problems. The average cloud cover for glaciers was 73.8% for MJJA and slightly higher for AMJJAS, or 74.4%. Monthly variability within the melt season was low with the highest values seen in April, July and September (78, 76 and 75%, respectively) and



**Table 2.** Comparison of MODIS albedo products (MOD10A1, MYD10A1, MCD43A1 and MCD11) with GAWS in-situ albedo on a monthly timescale.

|         | MOD10A1 | | MYD10A1 | | MCD43A3 | | MCD11 | |
|---------|------|-------|------|-------|------|-------|------|-------|
| Month   | RMSE | $R^2$ | RMSE | $R^2$ | RMSE | $R^2$ | RMSE | $R^2$ |
| May     | 8.66 | 0.82  | 8.34 | 0.84  | 8.28 | 0.84  | 7.9  | 0.86  |
| June    | 7.07 | 0.91  | 7.20 | 0.91  | 7.49 | 0.91  | 6.59 | 0.92  |
| July    | 7.08 | 0.92  | 6.30 | 0.93  | 7.09 | 0.91  | 6.49 | 0.93  |
| August  | 8.24 | 0.88  | 7.52 | 0.90  | 11.0 | 0.75  | 7.79 | 0.89  |

lower values in May, June, August and October (73, 73.5, 72.8 and 72.8% respectively, individual months are shown in Fig. B1–B3). The average highest cloud cover was observed for Eyjafjallajökull (80.3%), Drangajökull (79.6%), and Mýrdalsjökull (77%) for melt-season averages while the other glaciers have lower average cloud cover ranging from 71–74%.

The average daily cloud cover in MOD10A1 data was 79% and slightly lower for MYD10A1, or 78% based on data from
April to October each year for the period from 2000–2019. By joining these two products on a daily basis, cloud-obscured pixels were reduced to 74%. Temporal aggregation (11 days) of the products had an exponential decaying shape of unclassified pixel reduction with the highest benefit for aggregating 1 day. For this study, data were aggregated 5 days forward and backward allowing 11 days of both AQUA and TERRA MODIS albedo data to contribute to a daily average. This resulted in an average unclassified pixel reduction down to 12%.


The main advantage of the temporal aggregation of the data was the reduction of cloud-obscured pixels, which provides a more spatially continuous product in a simple and computationally efficient way. This comes with the primary disadvantages of response dampening of rapid changes, experienced as an smoothing effect on the albedo time series. This could pose a limitation on daily near real-time flow forecasting while for weekly to monthly time scale applications, the product should
be representative. Cloud detection in the MODIS products is based on the M*D35−L2 cloud mask providing four categories for discrimination of clouds, i.e. cloudy, uncertain, probably clear and confident clear. Cloud and snow confusion is known for MODIS data for many reasons, such as cold clouds with ice content, very similar spectral responses of some cloud types as snow, and cirrus clouds that are not detected (Sirguey et al., 2009; Box et al., 2012). The approach in this study to reduce cloud artifacts is based on robust statistics with a median-based outlier removal. The drawback of this approach is that with a
too strict criterion for rejection, valid data could be rejected, with loss of good quality data, especially in cases where surface albedo changes rapidly.



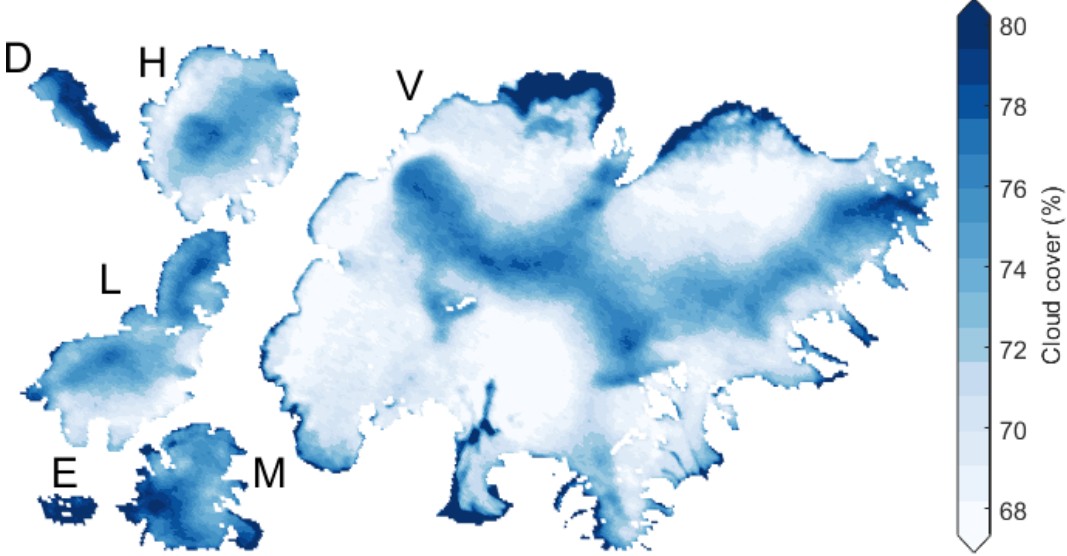

**Figure 3.** Average cloud cover for the main Icelandic ice caps for the extended melt season from May to September each year from 2000–2019 (average: 73.8%).

## 3.3 Annual and inter-annual variability of albedo

Inter-annual albedo variations for Icelandic glaciers were generally high. Figure 4 shows spatial patterns for melt-season mean albedo for the investigated glaciers for the period from 2000–2019 (MJJA). The lowest albedo values (<35%) were found in bare-ice areas where the winter snow cover generally is completely ablated during summer revealing dirty and impurity-rich bare ice. Higher albedo values (> 45–50%) were found in the accumulation areas associated with higher elevations and a shorter period of positive surface-energy balance during the melt season.

Figure 5 shows the average albedo distribution and relations to elevation in 100 m bands for the six largest ice caps and their sub-areas defined in Fig.1. Above 1500 m a.s.l. at Vatnajökull there were limited regional variability while more distinctive patterns were seen between the northern (NW and NE) and southern parts, especially in the southeast at lower elevations. In the southeast, the elevation of the glacier ranges all the way down to sea level while the glacier terminus was at a much higher elevation in the north (600–700 m a.s.l.). The average-albedo-elevation relationship for Vatnajökull, exhibits three elevation gradients. For elevations below 700 m a.s.l. the linear albedo gradient was ∼2.3%/100 m, ∼5.1%/100 m between 700–1300 and ∼0.5%/100 m for elevations above 1300 m. For Hofsjökull, the albedo was generally lower in the southeast than in the northern and southwest parts, the average albedo elevation gradient below 1400 m a.s.l. was 4%/100 m and 1.5%/100 m above 1400 m a.s.l. For Langjökull, the south and northeast areas had overall lower average albedo values compared with the north-western part of the glacier. At Langjökull, the albedo elevation gradient was 3.5%/100 m for the whole elevation range which was similar as for elevations below 1400 m a.s.l. at Hofsjökull, but note the start of a change towards a lower gradient at the higher elevations. The northwest part of Mýrdalsjökull had generally higher albedo compared to the southern part. The albedo





gradient is 3%/100 m while for the whole elevation range. Distinctive patterns were observed for the eastern and southern part of Drangajökull with lower average values for the south region. A very strong east/south cloud cover gradient was as well observed at Drangajökull (Fig. 3) that could explain these differences, indicating that less SW↓ reaches the surface accelerating the snow metamorphism and resulting in lower albedo. The average albedo elevation gradient was 3.0%/100 m for
Drangajökull and 2.7%/100 m for Eyjafjallajökull. In general, the southern parts of the main ice caps had lower albedo. This was most likely controlled or strongly influenced by orographic generation of precipitation in the dominating SE-SW wind providing more energy from rain and warmer temperatures at the surface, accelerating the snow metamorphism (Einarsson, 1984; Crochet et al., 2007; Björnsson et al., 2018).

Figure 6 shows the average distribution of albedo as a function of elevation bands (100 m intervals) and time for the period from 2000–2019. The annual maximum albedo value for all elevation bands was generally observed in early April associated with the last major snowfalls of winter. The lowest average albedo values were observed from mid July to mid August. For higher elevations (accumulation areas), the minimum values were associated with first snowfall which increases albedo. For bare-ice areas with impurity-rich ice, these impurities can be washed away from the glacier surface by rain which lead to higher
albedo without fresh snow, i.e. cleaner ice, with less impurities, in late summer.

Figure 7 shows average melt-season mean albedo for the glaciers and sub-areas defined in Fig.1. Glaciers were sorted from the highest to the lowest melt-season mean albedo for the whole analysis period (highest at the top of the figure), revealing certain spatial, temporal and feature position patterns. The lowest albedo values were observed for Mýrdalsjökull, Eyjafjallajökull
and Torfajökull, which all cluster together at the south coast of Iceland (Fig 1, box M). They were also all close to widespread unstable sandy surfaces subject to frequent high-velocity winds, driving numerous wind erosion events and dust production. These unstable erosive surfaces do not sustain seasonal snow cover far into the spring and summer, making them accessible for erosion earlier in the spring than similar areas in the north and east highlands near to Langjökull, Hofsjökull and Vatnajökull. These glaciers were relatively small as well, indicating that dust producing events can influence larger areas of the glaciers with
dust deposits. Following these glaciers on the south coast were smaller glaciers, with the exception of northwest Langjökull, with slightly higher annual average albedo. These were small alpine and valley glaciers with less elevation range and surface area compared with the large ice caps. The main ice-caps in Iceland, Vatnajökull, Hofsjökull and Langjökull had relatively high average albedo compared with the other glaciers with the exception of the northwestern part of Langjökull which was close to the Flosaskarð area known for extremely severer erosion (Arnalds et al., 2016). Drangajökull had the highest observed
albedo, its location was far from unstable surfaces that produce airborne dust and volcanic eruptions (2010, 2011) seem to have a minimal effect compared with other Icelandic glaciers. Albedo development at Drangajökull was likely mostly driven by snow metamorphism where snow grain size increases with time and energy input resulting in lowering of albedo.

On the temporal scale, various events influencing the melt-season mean albedo were observed in Fig. 7. For the south coast
glaciers (Fig 1, box M), the influence of 2010 volcanic eruption in Eyjafjallajökull and the post-eruption influence in 2011





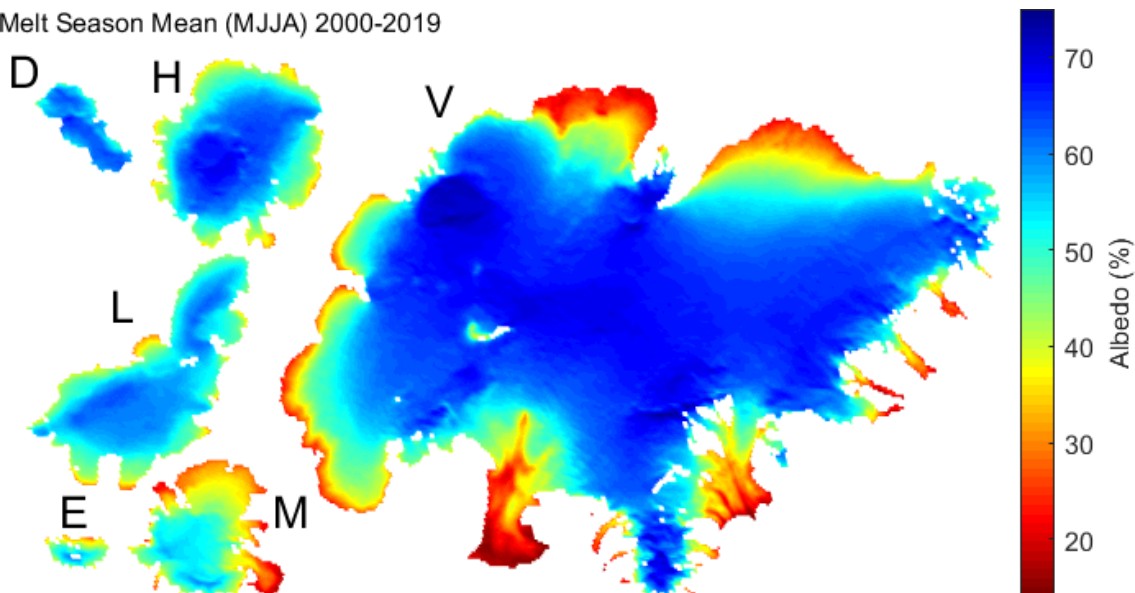

**Figure 4.** Spatial patterns of mean albedo for the period from 2000–2019 (MJJA). **D:** Drangajökull, **H:** Hofsjökull, **V:** Vatnajökull, **L:** Langjökull, **E:** Eyjafjallajökull and **M:** Mýrdalsjökull.

and 2012 were obvious, and there were also influence on other Icelandic glaciers, with the possible exception of Drangajökull, Hofsjökull Eystri, Snæfell and Norðurlandsjöklar in the north. The influence on albedo due to the 2011 volcanic eruption in Grímsvötn was seen in south west Vatnajökull. Generally albedo was lower for most glaciers in that year, excluding Drangajökull. In 2015, a cold spring and summer, with prolonged snow cover in the highlands, delayed the onset of melt, as well as

limiting the capabilities for airborne dust and tephra to be transported to the glacier surface. The highest melt-season mean albedo observed during the study period was in 2015 for all glacier, while the lowest albedo was seen in 2010. The melt season in 2019 was furthermore seen to be quite unique. Due to an early winter snow cover melt in the highlands in late April, the earliest and most extensive snow cover depletion for 20 years (MODIS period) (Gunnarsson et al., 2019), followed by a prolonged period with limited precipitation, great amounts of dust and sand from unstable sandy surfaces were transported to the

glaciers, providing Light Absorbing Particles that further enhance surface melt. Although similar singular events had been observed historically during the MODIS period, this development was observed at all Icelandic glaciers. Note must be taken when melt season average values are interpreted that they are influenced by the areal elevation distribution of each glacier or sub-area.

Seasonal variability of albedo for Icelandic glaciers was generally high. Figure 8 shows glacier average seasonal albedo

distribution for 2000–2019 plotted with selected years for Vatnajökull, Hofsjökull, Langjökull, Mýrdalsjökull, Eyjafjallajökull and Drangajökull. The average albedo generally declines from the maximum observed in the first two weeks of April each year (70–80%) to an annual minimum in August. The average minimum observed value is 40–45% for Vatnajökull, Hofsjökull, Langjökull and Drangajökull but reaches lower values at Mýrdals- and Eyjafjallajökull (>30%). Glacier runoff generally





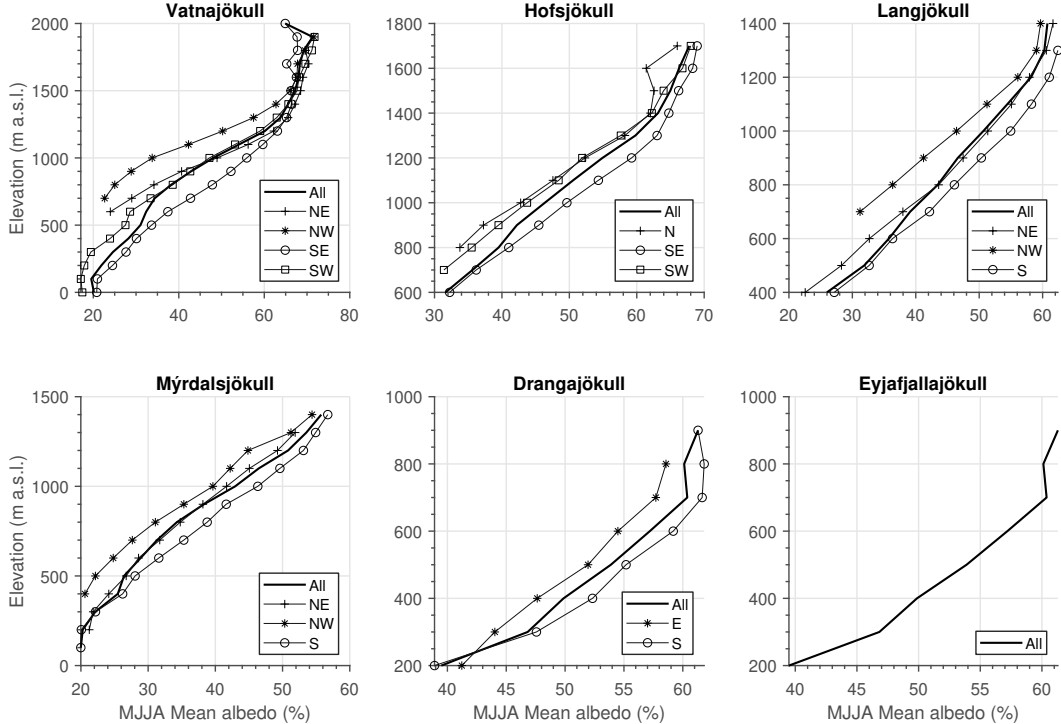

**Figure 5.** Average albedo for the period as function of elevation for the period 2000–2019. Data are shown for the six largest ice caps for the whole glaciers (All) as well as for the sub-areas defined in Fig. 1. Note: the elevation range varies between figure axes (y-axis).

peaks in late June and July (midsummer) (Schmidt et al., 2018) with low albedo and maximum incoming shortwave irradinace

near the summer solstice. The variability similarly gradually increased in June, July and August and was generally highest in August. In the fall, seasonal weather patterns in Iceland shift with lowering temperatures and an increase in precipitation following shorter days due to a gradual increase in solar zenith angles (Einarsson, 1984; Hanna et al., 2004; Björnsson et al., 2007; Björnsson et al., 2018). Frequently in the latter half of August and beginning of September, the first snowfall is observed to increase albedo with fresh highly reflective snow. It was not uncommon to see the albedo lower again after the first snowfall

due to liquid precipitation or other events that melt the fresh snow cover over the bare glacier ice. This affects the variance of albedo in August and September.

Figure 8 also shows how albedo develops through the melt season for selected abnormal years. The influence of explosive volcanic eruptions in Grímsvötn in Vatnajökull are shown in 2005 (erupted in November 2004) and 2011 and the Eyjafjal-

lajökull eruption in 2010. These events generally influence the albedo of Icelandic glaciers as tephra is discharged into the atmosphere and transported by wind over wide areas. In 2015, seasonal mass balance programs for Vatnajökull, Langjökull and Hofsjökull reported unusually thick winter snow cover followed by a cold and cloudy spring and summer which resulted in a positive net surface mass balance, for the first time in 20 years (Pálsson et al., 2016, 2017; Þorsteinsson et al., 2017). Figure





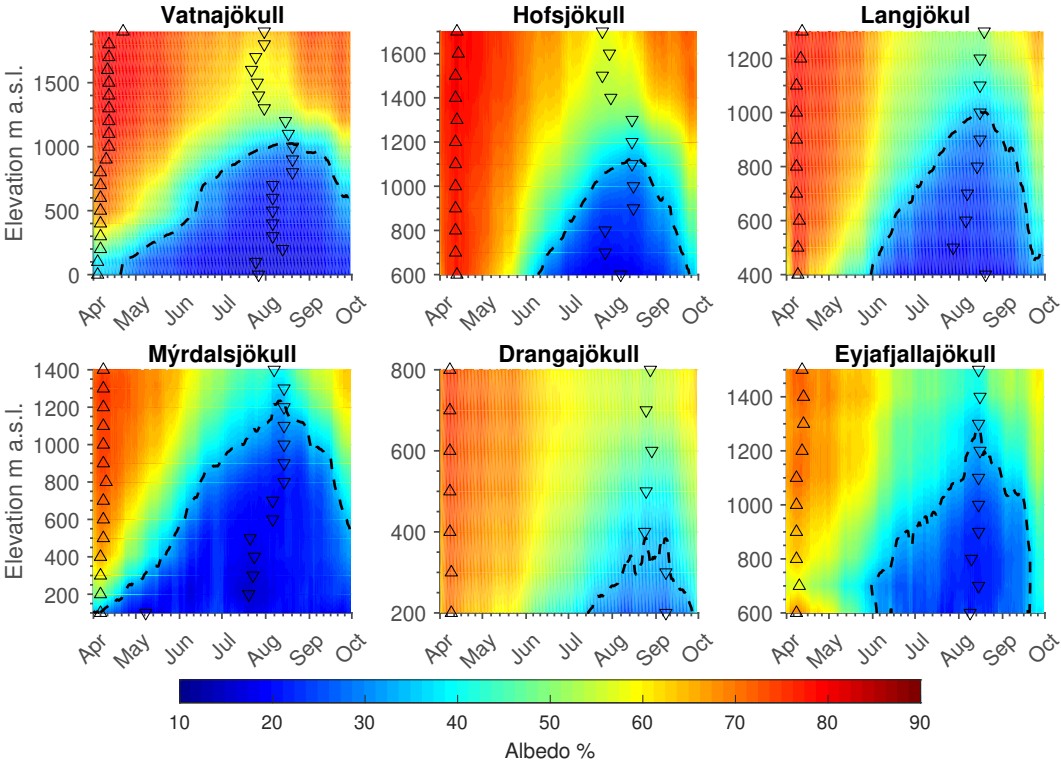

**Figure 6.** Albedo as a function of elevation and time for the period 2000–2019. Triangles show the max/min values associated with each elevation band and the dotted black line shows the isoline for 34% albedo, as defined by Cuffey and Paterson (2010) for bare glacier ice.

8 shows the development of albedo in 2015 as the highest average values for the study period.


Figure 9 shows the spatial distribution of seasonal average albedo as anomalies from the mean. Blue colors represent anomalies above the mean, i.e. higher albedo values while red areas represent values below the mean. Decisive negative patterns were observed in 2010 and 2011. These relate to the volcanic eruptions in Eyjafjallajökull (2010) and Grímsvötn (2011) as tephra dispersal from explosive eruptions produces high volumes of airborne tephra (Gudmundsson et al., 1997; Guðmundsson et al.,

2012; Tesche et al., 2012; Liu et al., 2014). Airborne tephra can be transported by high plumes that can extend several kilometres into the atmosphere and be transported great distances, up to several hundred kilometers (Guðmundsson et al., 2012; Watson et al., 2016). Tephra dispersal and fallout patterns from explosive eruptions depend on multiple factors, including plume height, particle size distribution, and wind direction and velocity among various geological factors. No eruption occurred in 2012 but residual effects were observed as ash deposits from previous eruptions were carried with the prevailing wind direc-

tions, enhancing melt due to the lowering of albedo. These effects were most clear for Eyjafjallajökull and Mýrdalsjökull but also contribute to negative anomalies for Vatnajökull. The impact of dust deposition on albedo in 2012 for Vatnajökull was investigated by Wittmann et al. (2017a) using dust-mobilization models to calculate dust emission and a dispersion model to





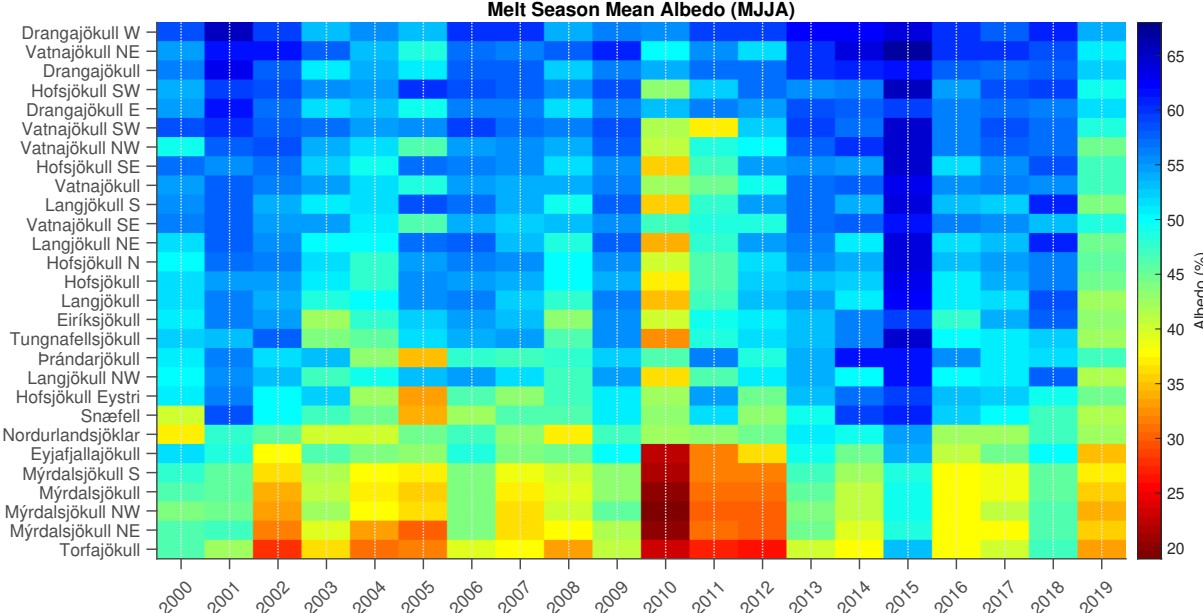

**Figure 7.** Average melt-season albedo for the studied glaciers. The glaciers are sorted from the lowest 2000–2019 melt season average albedo to the highest. For the larger glaciers, data are provided for individual ice-flow basins, see Figure 1.

simulate atmospheric dust dispersion and deposition on the glacier surface. The main conclusion was that the influence of dust on albedo could lead up to 40% melt increase which confirms the influence of these events on seasonal glacier melt.


Another influencing factor for negative albedo anomalies was dust, sand and other Light Absorbing Particles (LAP) transported from the proglacial areas and sandy deserts which cover more than 22% of Iceland (Arnalds et al., 2016; Wittmann et al., 2017a). Plume shape patterns could be identified especially for the northern part of Vatnajökull indicating airborne LAP deposits on the glacier surface. As an example, in 2001, 2003, 2007, 2008 and 2013, such patterns were observed in the north-

ern part of Vatnajökull (Brúarjökull glacier outlet) extending from the Kverkfjöll mountain range high in the accumulation area as local negative albedo anomalies. These were unlikely to be linked to local climatology resulting in such distinctive anomalies as such events or dominating patterns would influence larger areas. In 2014–15, the lava flow field of the Holuhraun non-explosive eruption covered about 84 km$^2$ of volcaniclastic sandy desert and proglacial areas north of Vatnajökull. Since then, similar plume shaped albedo anomalies were not observed in the data. It is probable that the extent of the lava flow field

reduces the dust production of this area significantly, although this cannot be quantified at this point in time, more data over a range of climatologies are needed to fully understand the impact of the Holuhraun eruption on dust production. Figure 9 also shows an interesting anomaly pattern for 2019. All the major ice caps had largely negative anomalies driven by dust and mineral deposits with an early onset in the spring. The events leading up to these anomalies have already been discussed above. In 2000, large negative anomalies were seen in Dyngjujökull and Brúarjökull (Northern Vatnajökull). These are unlikely linked





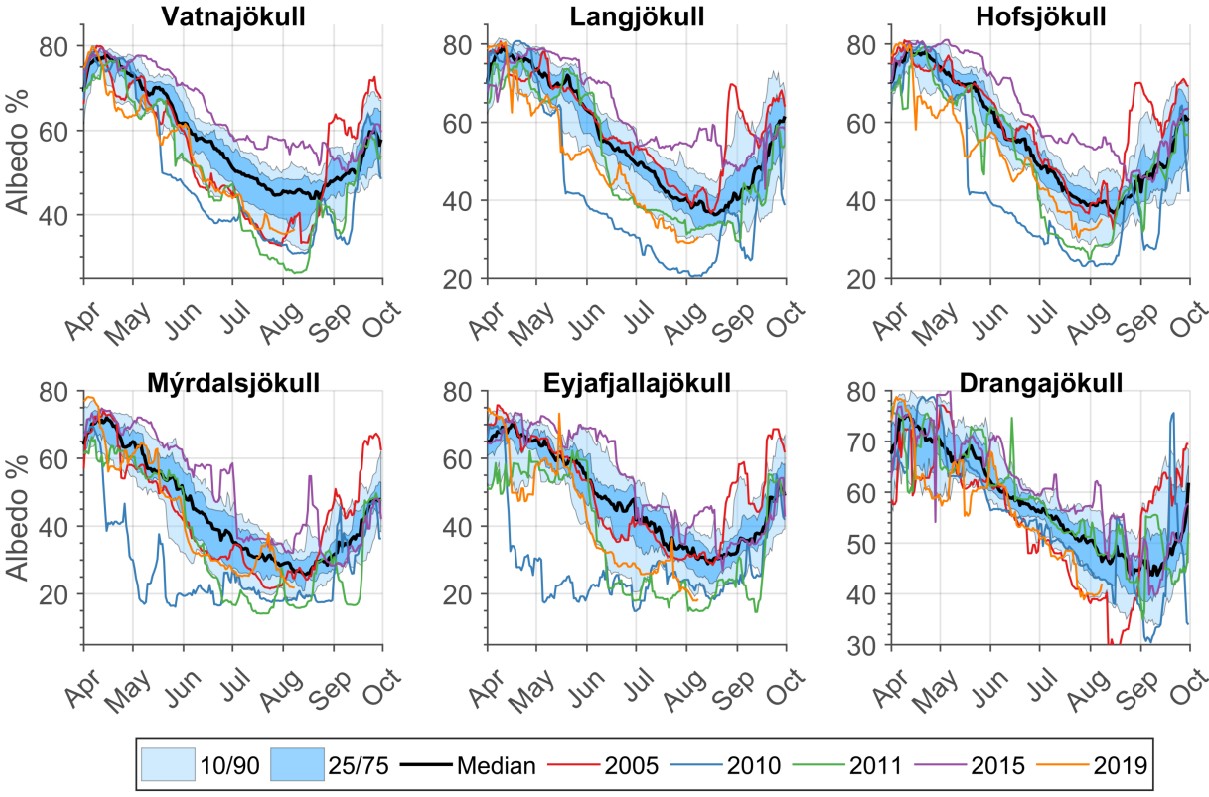

**Figure 8.** Seasonal variations of average albedo for selected Icelandic glaciers from the MCD11 product for April to October for the from 2000–2019.

to the 1999 Hekla eruption and are presumed a combination of residual effects from the Gjálp eruption in 1996 and dust transported from the pro glacial areas near the glacier terminus. Landsat images from the summer in 2000 show the surroundings near Gjálp covered in tephra as a possible dust source in a combination with pro glacial areas.





**Figure 9.** Spatial patterns for albedo anomalies for the period from 2000–2019.

## 3.4 Trends of albedo

Figure 10 shows annual spatial patterns of albedo trends. For Vatnajökull, negative albedo melt season trends were found in the lowest areas of the glacier with the exception of the northwestern part (Dyngjujökull). Negative trends at the terminus of glaciers were expected due to steady glacier retreat for the past decades with an associated debris deposits on dead-ice (Einarsson, 2018). In general, negative trends extend farther into the accumulation area in the southwest while a growing positive trend was observed in the upper part of the ablation area in the northern part with the exception of the Brúarjökull termini area. Positive trends in the upper part of the ablation area in the northern part (Brúarjökull and Dyngjujökull) of the ice-cap are significant over most of the area. These positive melt season trends were seen for the area near the equilibrium line elevation





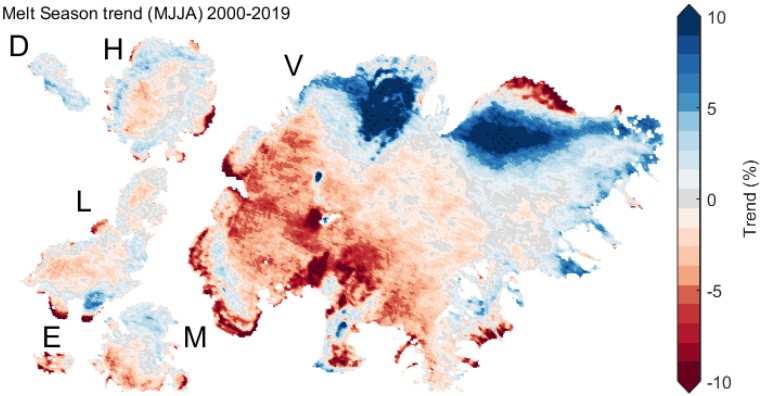

**Figure 10.** Spatial patterns for albedo trends during the melt season (MJJA) for the period from 2000–2019.

at Hofsjökull, for most of the extent of Drangajökull, in the northern area of Mýrdalsjökull and distributed parts of Langjökull with the exception of Eyjafjallajökull, suggesting a trend towards either increased snowfall or increased snow melt at these glacier outlets. As a melt-season average trend (Fig. 10) these positive trends are only significant in the ablation area in the northern part while negative trends were identified at many glacier termini, due to the steady glacier retreat in the past decades, reduction in the duration of snow cover over low-albedo bare ice while for the accumulation area in southwest Vatnajökull, the trend is strongly controlled by volcanic ash fallout in 2010 and 2011.

Figure 11 shows average monthly-mean albedo for the main ice caps for the study period, the associated linear trends and the average linear slope of the trend. For all the glaciers with the exception of Drangajökull in June, the average linear slope for May and June was negative, i.e. lower average albedo earlier in the spring. For Vatnajökull, Hofsjökull and Langjökull, the trend was strongly influenced by low May and June albedo in 2017 and 2019. These trends indicate that an more incoming shortwave energy is absorbed at the surface during these months with lower albedo. In July and August, the trend was in general positive, trending towards higher mean albedo. The trends were only statistically significant in July and August for Drangajökull and in July for Hofsjökull. Positive trends could indicate more extensive or earlier snowfall in July and August with fresh highly reflecting snow.





**Figure 11.** Average monthly mean albedo for the main ice caps in Figure 8. The mean, standard deviation and trend (Δy) are shown. Linear trend detremined from all years is shown with black lines while red lines exclude the 2010 and 2011 data to omit the influence of volcanic tephra and ash.



## 4 Conclusions

In this study, a gap-filled satellite-observed albedo dataset for Icelandic glaciers (MCD11) was produced from daily MODIS Aqua and Terra observations from early 2000 until 2019 at a 500 m spatial resolution. Overall, good visual and statistical agreement was found between the MCD11 data and in-situ albedo from GAWS observations over a range of elevations and glacier locations. Overall, higher RMS errors were found in the ablation zone which could be related to higher albedo variability within a MODIS pixel for impurity-rich bare ice in the ablation zone, indicating that care must be taken when comparing point-based in-situ observations with data with larger spatial footprint.

The main results show that the large seasonal and inter-annual variability in surface albedo for Icelandic glaciers was captured by the MCD11 data although limited in-situ data were available for the smaller glaciers. Icelandic glacier albedo was observed to be influenced by variability in climate, tephra deposits from volcanic eruptions, and airborne dust from widespread unstable sandy surfaces which are subject to frequent wind erosion and dust production.

Although not directly quantified in this study, it was clear that Light Absorbing Particles were a major contributor to annual glacier melt through additional radiative forcing at the surface. Light Absorbing Particles originate both from airborne dust sourced outside of the glacier, from volcanic eruptions, as well as from residual effects several years after eruptions. This illustrates the importance of a correct representation of surface albedo for glaciers in Iceland, as surface albedo is a dominating control on mass balance. Therefore, caution must be exercised in applying conventional albedo parametrization in hydro- and glaciological model, which often estimate albedo based on temperature, precipitation and time, especially for short and seasonal forecasting on catchment scales.

Significant negative albedo trends over the study period were found in northern Vatnajökull while other areas and glaciers have a glacier wide non-significant trend. Average linear trends for monthly data indicate that albedo generally decreased over the study period in May and June whereas a general albedo increase was observed in July and August, although, statistically non-significant in all cases with the exception of Hofsjökull in July and Drangajökull in July and August.

The incorporation of the MCD11 albedo product provides capabilities to improve surface mass balance and streamflow forecasting from glaciers. In the case of future volcanic eruptions, the presented methodology allows for rapid assessment of glacier albedo changes in near-real-time and the associated influence on melt which has a direct impact on hydropower production in Iceland and possibly civil infrastructure in some cases. A limitation related to estimating the impact of tephra fallout on a glacier surface from optical data is the assessment of tephra thickness, as very low observed albedo could indicate melt increase due to more surface energy absorbed by the surface but could as well indicate an isolating layer limiting melt due to a thick tephra layer.

Finally, it is noted that the methodology applied in the study, based on MODIS data, can be applied to other satellite albedo products, such as Sentinel 3, to extend the temporal range beyond the MODIS mission, allowing for short-term as well as long-term monitoring of albedo variations for glaciers in Iceland.

*Code and data availability.* Code used in the project to process data is available at https://github.com/andrigunn/aig2. MODIS data are
465 available from https://nsidc.org/data. Geospatial data for Iceland are available from the National Land Survey of Iceland at https://atlas.lmi.is. Glacier automatic weather station data is available upon request.

*Author contributions.* AG conceived and designed the study, performed the analyses, and prepared the manuscript. SMG contributed to the study design, interpretation of the results, and writing of the manuscript. FP, TJ and ÓGBS contributed to the interpretation of the results and writing and reviewing of the manuscript.

470 *Competing interests.* The authors declare that they have no conflict of interest.

*Disclaimer.* TEXT



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





**Appendix B: Supplement material**

Supplement material. Will be processed to a separate file during final processing



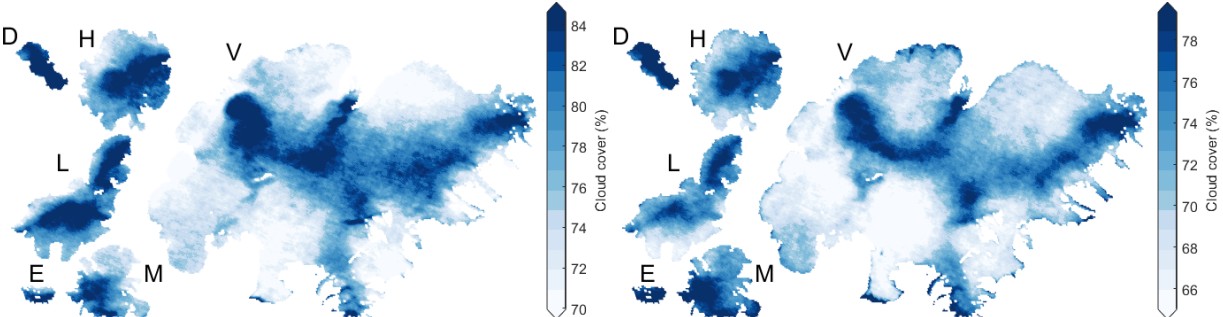

**Figure B1.** Monthly average cloud cover for selected glaciers in Iceland in April (left) and May (rigth).

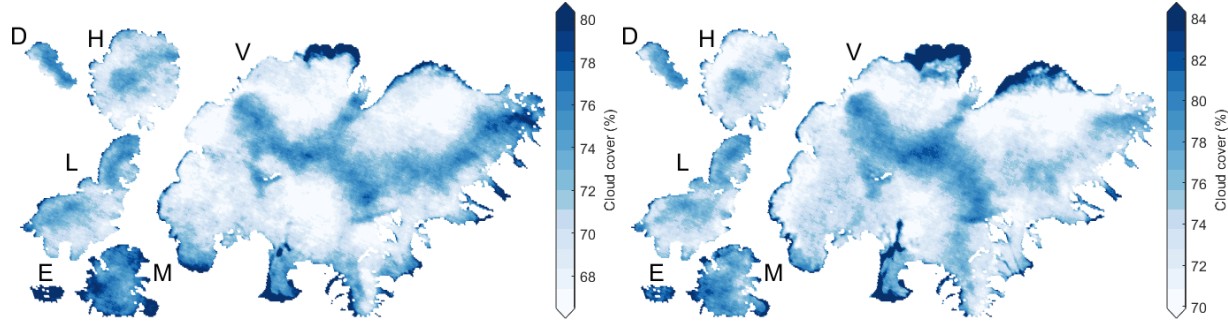

**Figure B2.** Monthly average cloud cover for selected glaciers in Iceland in June (left) and July (right).

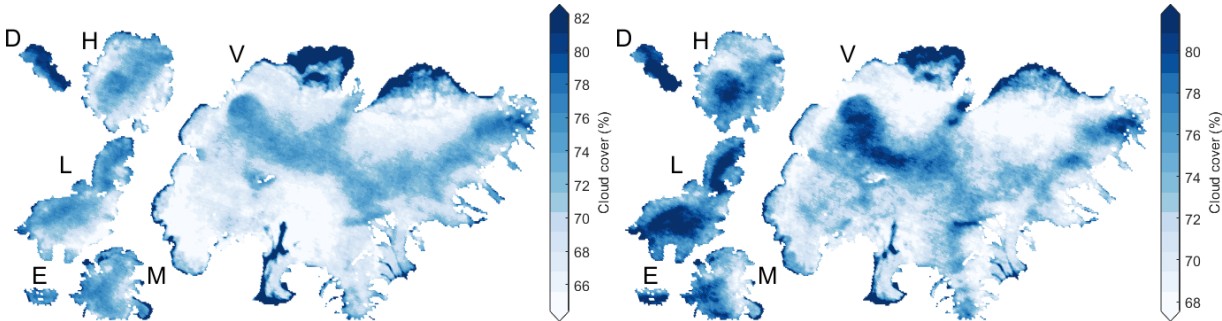

**Figure B3.** Monthly average cloud cover for selected glaciers in Iceland in August (left) and September (right).



**Table B1.** RMS error, $R^2$ values and number of months of overlapping data (n) for individual station comparison on a monthly time scale for MOD10A1, MYD10A1, MCD43A and MCD11.

| Station | MOD10A1 | | | MYD10A1 | | | MCD43A3 | | | MCD11 | | |
|---|---|---|---|---|---|---|---|---|---|---|---|---|
| | RMSE | R2 | n | RMSE | R2 | n | RMSE | R2 | n | RMSE | R2 | n |
| Kokv | 9.78 | 0.74 | 5 | 0 | - | 0 | 9.25 | 0.77 | 5 | 9.42 | 0.77 | 6 |
| BRE | 10.02 | 0.31 | 101 | 9.84 | 0.37 | 93 | 10.95 | 0.31 | 80 | 10.54 | 0.40 | 109 |
| B10 | 10.67 | 0.21 | 102 | 11.30 | 0.17 | 93 | 11.56 | 0.12 | 84 | 11.69 | 0.26 | 109 |
| B13 | 13.24 | 0.37 | 107 | 11.59 | 0.48 | 93 | 9.83 | 0.61 | 76 | 13.69 | 0.36 | 108 |
| B16 | 8.02 | 0.39 | 102 | 5.14 | 0.62 | 94 | 5.46 | 0.59 | 26 | 10.63 | 0.13 | 105 |
| BRE1 | 9.16 | 0.42 | 102 | 9.39 | 0.43 | 93 | 10.49 | 0.34 | 97 | 9.90 | 0.47 | 109 |
| BRE4 | 7.68 | 0.85 | 33 | 9.87 | 0.76 | 34 | 10.90 | 0.75 | 34 | 8.50 | 0.85 | 36 |
| BRE7 | 7.58 | 0.20 | 17 | 6.63 | 0.39 | 17 | 7.23 | 0.23 | 11 | 6.16 | 0.47 | 17 |
| T01 | 16.76 | 0.45 | 18 | 13.92 | 0.59 | 12 | 12.60 | 0.76 | 20 | 9.53 | 0.86 | 20 |
| T03 | 11.84 | 0.67 | 99 | 10.54 | 0.73 | 86 | 13.55 | 0.59 | 98 | 12.67 | 0.64 | 102 |
| T06 | 10.27 | 0.53 | 98 | 14.69 | 0.26 | 87 | 7.11 | 0.73 | 69 | 13.91 | 0.26 | 100 |
| L01 | 12.41 | 0.73 | 95 | 13.26 | 0.69 | 87 | 10.24 | 0.84 | 99 | 11.43 | 0.80 | 103 |
| L05 | 8.35 | 0.71 | 100 | 8.58 | 0.69 | 92 | 7.93 | 0.75 | 106 | 9.83 | 0.65 | 114 |
| K06 | 18.06 | 0.003 | 35 | 17.85 | 0.02 | 35 | 21.57 | 0.03 | 11 | 19.37 | 0.08 | 36 |
| MYRA | 9.08 | 0.55 | 20 | 9.56 | 0.51 | 20 | 18.68 | 0.02 | 16 | 18.53 | 0.019 | 21 |
| HSA09 | 5.75 | 0.93 | 11 | 9.27 | 0.83 | 11 | 5.67 | 0.94 | 12 | 5.18 | 0.96 | 13 |
| HSA13 | 5.81 | 0.74 | 11 | 4.05 | 0.87 | 11 | 5.59 | 0.78 | 11 | 5.47 | 0.79 | 13 |
| SKE02 | 6.00 | 0.0004 | 3 | 0.25 | 0.99 | 3 | 2.07 | 0.90 | 3 | 0.12 | 0.99 | 3 |
| Hof01 | 14.99 | 0.14 | 63 | 15.31 | 0.12 | 61 | 5.21 | 0.82 | 24 | 19.18 | 1.1e-06 | 66 |
| Hosp | 10.56 | 0.27 | 53 | 10.59 | 0.26 | 53 | 10.96 | 0.27 | 56 | 11.39 | 0.22 | 58 |





**Figure B4.** Albedo comparison results from monthly averaged MODIS data for May, June, July and August for the period from 2000 - 2019 where data were available for MOD10A1, MYD10A1, MCD43 and MCD11 data products.