# Peer review of "Annual and inter-annual variability and trends of albedo of Icelandic glaciers"

_The Cryosphere, 2019_

## Referee Comment (RC1) · Simon Gascoin (Referee) · 29 Feb 2020

This paper presents the application of MODIS snow albedo products to characterize the spatial-temporal variability and trends of glacier albedo in Iceland. The albedo data are derived from the M*D10A1 products after interpolating missing data due to the frequent cloud cover. The topic is interesting since Icelandic glaciers are frequently exposed to volcanic ashes deposition causing large albedo changes and thereby modulating their response to climate forcing (Schmidt et al. 2017). A strength of the study is the extensive in situ dataset that was used to evaluate the MODIS products (20 AWS). My main concern regarding this study is the apparent lack of novelty with respect to previous works by Möller et al. (2014) and Gascoin et al. (2017) who also studied the albedo changes over Icelandic glaciers. Some figures in the manuscript provide similar

information as Gascoin et al. (2017).

In particular the introduction does not clearly state why it was needed to go beyond previous studies by Möller et al. (2014) and Gascoin et al. (2017). I see some differences that could indeed justify this new study. The authors used M*D10A1, while the latter studies used MCD43A3. However, the authors should strengthen this part of the manuscript by providing a detailed comparison of both products. As it stands, the results cannot be compared to those reported by Gascoin et al. (2017) mainly because the authors computed the RMSE and correlations at the monthly time step whereas we used daily values (see L245 "The comparison presented here is in fact similar to previous work on Icelandic glaciers by Gascoin et al. (2017) where the MCD43A3 was evaluated with RMS errors ranging from 8–21%."). Looking at Tab B1 it seems that M*D10 products are more accurate than MCD43?

Also, an important aspect is that MCD43 provides albedo over all land masses, whereas M*D10A1 provides only albedo of the pixels that are detected as snow-covered. This can be an issue in Iceland where large regions of glaciers may not be detected as snow due to the tephra layer. This issue should be investigated to make sure that the MCD11 product is not interpolating the albedo of clear-sky, snow-free pixels.

The trends should be masked or marked where MK test is not significant (Figure 10).

The improvements in MCD11 albedo with respect to the original product are very small (about 0.01 RMSE, Tab.2). In addition it is indicated (L181) that the thresholds for outliers rejection were manually adjusted so the conclusions remain limited to this study. The main benefit of MCD11 is rather that it is a gap-free product which facilitates the utilization of the data.

The authors indicate that a motivation of their work is the integration of this albedo product in operational snow melt runoff model. It would be useful to have more background information on this aspect. What albedo is currently used by Landsvirkjun or

other agencies? Is the developed product compliant with operational context if there is a lag of at least 5 days before updating the albedo (since temporal interpolation is based on a 10 days window)?

Minor comments

L31 in an maritime climate

L34: Seasonal glacier melt : what does it mean: seasonal snow and ice melt from the glacier area

L41: are

L93: this paragraph gives me the impression to come out of the blue. The objective should be more clearly linked to the literature review and identified knowledge gaps.

L153: "Daily averages" is not the correct wording if it refers to of hourly albedo values. I understand from the above paragraph that the daily albedo was in fact calculated from daily sums of incoming and reflected radiation (which is recommended to reduce measurement noise).

L168: what is a "median based statistical rejection of outliers."

L173: I don't think you need these references to justify this general statement.

L184: these pixels are not unclassified, since they are classified as cloud.

L185: this approach is very similar to our algorithm for cloud pixels interpolation in MOD10 products (Gascoin et al. 2015). We used the same predictors. It should be cited if it has inspired your own algorithm.

L188 Correspondingly reads a bit odd here

L191 "monthly, basis"

L204: The calculations were

L215-220 the whole paragraph should be removed (it is method, not results)

L246: results are not directly comparable (daily vs. monthly) (see my main comments)

L253: "indicating high sub-pixel albedo variability" This is a bit vague and unexpected comment since large areas of Icelandic ice caps have a rather homogeneous surface (in comparison with Alpine glaciers for example). We studied albedo subpixel variability from Landsat data to explain the discrepancy between AWS measurements and MODIS retrieval. L273 experienced as an smoothing

Fig 3: a similar figure can be found in Gascoin et al 2017

Fig 4, 6, 7: rainbow colormaps are not recommended (see e.g. https://www.nature.com/articles/519291d)

Fig 6: the figure does not display correctly on my computer, I suggest to replace it by a bitmap (raster) version

L440: this sentence should be removed or reformulated since there is no information on glacier mass balance in this study

L462: Do you mean when MODIS will stop operating? Note that the successor of MODIS is rather VIIRS.

---

## Referee Comment (RC2) · Pavla Dagsson Waldhauserova (Referee) · 26 May 2020

The authors provide well-written detailed study on albedo changes of all Icelandic major glaciers using a comparison of MODIS snow albedo products and in situ measurements. This study could also serve as a comprehensive review of rapidly changing glaciers in Iceland with focus on impacts on their changing albedo. It brings insights into albedo analysis in problematic cloud-obscured region while providing novel findings on linear albedo gradients and dust plume shape patterns on snow and ice. Direct impacts of explosive volcanic eruptions as well as severe and moderate dust storms on the glaciers are evaluated. Additionally, possible indirect impacts of effusive eruptions such as Holuhraun 2014-2015 are suggested. It is clear that the authors know perfectly the local environment and its past. The data from the MODIS products were

carefully screened during extensive manual quality control and results were evaluated with valuable data from in situ ICE-GAWS network. The greatest contribution of this study is that the data set does not only include major explosive eruptions and cold years, but it includes extremely rare year 2019, dry and dusty in the southern part of Iceland. This allows the authors to compare the impacts of volcanic ash and general volcanic/glacial dust on the albedo at the same level. There are minor errors in references that should be stated in ascending order and several references could be added. I would recommend publication after minor revisions.

Specific comments:

L18-95 – Introduction

Consider to add studies on snow albedo reductions due to volcanic dust, eg. Meinander et al., 2014, Peltoniemi et al., 2015, Dagsson-Waldhauserova et al., 2015, Zubko et al., 2019).

Kylling et al., 2018 calculated the instantaneous radiative forcing of the bottom of the atmosphere due to mineral dust deposited on snow as 0.135 W m-2.

Kylling A., Groot Zwaaftink, C. D., Stohl, A., 2018. Mineral dust instantaneous radiative forcing in the Arctic. Geophysical Research Letters, 45. doi: 10.1029/2018GL077346.

Peltoniemi, J. I., Gritsevich, M., Hakala, T., Dagsson-Waldhauserová, P., Arnalds, Ó., Anttila, K., Hannula, H.-R., Kivekäs, N., Lihavainen, H., Meinander, O., Svensson, J., Virkkula, A., de Leeuw, G., 2015. Soot on snow experiment: bidirectional reflectance factor measurements of contaminated snow. The Cryosphere 9, 3075-3111.

Dagsson-Waldhauserova, P., Arnalds, O., Olafsson, H., Hladil, J., Skala, R., Navratil, T., Chadimova, L., Meinander, O., 2015. Snow-dust storm A case study from Iceland, March 7th 2013. Aeolian Research 16, 69–74.

Meinander, O., Kontu, A., Virkkula, A., Arola, A., Backman, L., Dagsson-Waldhauserová, P., Järvinen, O., Manninen, T., Svensson, J., de Leeuw, G., and Leppäranta, M., 2014. Brief Communication: Light-absorbing impurities can reduce the density of melting snow. The Cryosphere 8, 991-995.

Zubko, N., Muñoz, O., Zubko, E., Gritsevich, M., Escobar-Cerezo, J., and Berg, J., 2019. Light scattering from volcanic-sand particles in deposited and aerosol form. Atmos. Env. 215, 116813. doi: 10.1016/j.atmosenv.2019.06.051

L40 and L85 – remove 'a' in Wittmann et al., 2017a. Consider to add Gascoin at al., 2017 here.

L40-41 – 'surface albedo IS the dominating factors' - change ARE->IS, FACTORS->FACTOR

L44-45 – ..but it IS limited...

L47 – Stroeve et al. 2001? As in reference, not 2002.

L70-72 – Can you please rephrase the sentence or cut into two sentences. It is difficult to understand.

L144 – Van Den Broeke et al., 2004 a,b?

L164 – Table 1 – What do you mean by 'average location'?

L192-193 – opening brackets are missing

L229-230 – Do you mean annual melt rates here?

L230 – Sand particles have certain size resolution, maybe 'dust' is better here. Or 'sand and dust'.

L253– Small scale spatial variability of albedo could be also discussed here. See Hartl et al., 2020.

Hartl, L., Felbauer, L., Schwaizer, G., and Fischer, A.: Small scale spatial variability of bare-ice albedo at Jamtalferner, Austria, The Cryosphere Discuss., https://doi.org/10.5194/tc-2020-92, in review, 2020.

L289-308 – Linear albedo gradients are really important and well discussed here. However, the role of local impurities should be also mentioned here. General lower albedo values at certain parts of Hofsjökull, Langjökull and Myrdalsjökull coincides well with location of dust source areas described in Arnalds et al., 2016, and classified as severe or extremely severe erosion areas. This should be also included here in the discussion. There is also work from Antarctica showing the vertical gradient of local dust impurities on glacier that could be discussed here. See Kavan et al., 2020.

Kavan, J., Nyvlt, D., Láska, K., Engel, Z., and Knazková, M. (2020) High latitude dust deposition in snow on the glaciers of James Ross Island, Antarctica. Earth Surf. Process. Landforms, https://doi.org/10.1002/esp.4831

L319-321 – General trends in annual albedo (lowest values vs. highest values) correspond to the long-term dust storm frequency studies in Iceland. For evaluation, consider these three studies:

Nakashima, M. and Dagsson-Waldhauserová, P., 2019. A 60 Year Examination of Dust Day Activity and Its Contributing Factors From Ten Icelandic Weather Stations From 1950 to 2009. Frontiers in Earth Science 6, 245-252. DOI:10.3389/feart.2018.00245

Butwin, M.K., von Löwis, S., Pfeffer, M., and Thorsteinsson, Th., 2019. The Effects of Volcanic Eruptions on the Frequency of Particulate Matter Suspension Events in Iceland. Journal of Aerosol Science 128, 99-113.

Dagsson-Waldhauserova, P., Arnalds, O., Olafsson, H., 2014. Long-term variability of dust events in Iceland. Atmospheric Chemistry and Physics 14, 13411-13422. DOI:10.5194/acp-14-13411-2014.

L322-323 – Such unstable erosive surfaces are defined as 'dust hot spots' and it has been shown that dust events occur frequently in southern parts of Iceland in winter. Examples here:

Dagsson-Waldhauserova, P., Arnalds, O., Olafsson, H., 2014. Long-term variability of dust events in Iceland. Atmospheric Chemistry and Physics 14, 13411-13422. DOI:10.5194/acp-14-13411-2014

Dagsson-Waldhauserova, P., Renard, J.-B., Olafsson, H., Vignelles, D., Berthet, G., Verdier, N., Duverger, V., 2019. Vertical distribution of aerosols in dust storms during the Arctic winter. Scientific Reports 6, 1-11.

Dagsson-Waldhauserova, P., Arnalds, O., Olafsson, H., Hladil, J., Skala, R., Navratil, T., Chadimova, L., Meinander, O., 2015. Snow-dust storm A case study from Iceland, March 7th 2013. Aeolian Research 16, 69–74.

L329 – delete 'r' in severer

L329-332 – Just to comment. There are few cases when Drangajökull and Westfjords receive dust from the dust hot spots in central and South Iceland. Such events were captured by satellite or by dust model frequently in 2019.

L353 – '>30%' Did you mean <30%

L375 – Liu et al. (2014) do not really refer to volcanic ash from eruption, but dust event with maybe some relicts of ash. Their sample was collected in 2013 and they describe a dust event in 2013. I would suggest to remove this from the references here.

L380-381 – It was also induced by high dust storm activity in that area, see Möller et al., 2019. Volcanic ash is usually being removed fast from surfaces in Iceland, in < 1 year. See Butwin et al., 2019 or Arnalds et al., 2013.

Butwin, M.K., von Löwis, S., Pfeffer, M., and Thorsteinsson, Th., 2019. The Effects of Volcanic Eruptions on the Frequency of Particulate Matter Suspension Events in Iceland. Journal of Aerosol Science 128, 99-113.

Arnalds, O., Thorarinsdottir, E.F., Thorsson, J., Dagsson-Waldhauserova, P., Agustsdottir, A.M., 2013. An extreme wind erosion event of the fresh Eyjafjallajokull 2010 volcanic ash. Nature Scientific Reports 3, 1257.

**TCD**

Interactive
comment

[Figure]

L382 – Wittmann et al. (2017a). Why 'a'?

Figure 8 – Correct the title – delete 'for the'?

L400 – 1999 Hekla – Are you talking about 26th Feb 2000 Hekla eruption here?

L386-398 – When discussing dust influence on the albedos, you can also include that not only volcanic ash can be lifted to high altitudes and transported long distances. It is also Icelandic volcanic dust that can reach several km heights and travel long distances of thousands of km:

Dagsson-Waldhauserova, P., Renard, J.-B., Olafsson, H., Vignelles, D., Berthet, G., Verdier, N., Duverger, V., 2019. Vertical distribution of aerosols in dust storms during the Arctic winter. Scientific Reports 6, 1-11.

Djordjević D., Tošić I., Sakan S., Petrović S., ĂŘuričić-Milanković J., Finger D.C. and Dagsson-Waldhauserová P. 2019. Can Volcanic Dust Suspended From Surface Soil and Deserts of Iceland Be Transferred to Central Balkan Similarly to African Dust (Sahara)? Frontiers in Earth Sciences 7, 142-154.

Moroni B., Ólafur Arnalds, Pavla Dagsson Waldhauserová, Crocchianti, S., Vivani R., and Cappelletti, D. 2018. Mineralogical and chemical records of Icelandic dust sources upon Ny-Ålesund (Svalbard Islands). Frontiers in Earth Science 6, 187-219.

Beckett, F., Kylling, A., SigurĂřardóttir, G., von Löwis, S., and Witham, C., 2017. Quantifying the mass loading of particles in an ash cloud remobilized from tephra deposits on Iceland, Atmos. Chem. Phys., 17, 4401-4418.

Ovadnevaite J., Ceburnis D., Plauskaite-Sukiene K., Modini R., Dupuy R., Rimselyte I., Ramonet R., Kvietkus K., Ristovski Z., Berresheim H., O'Dowd C.D., 2009. Volcanic sulphate and arctic dust plumes over the North Atlantic Ocean. Atmospheric Environment 43, 4968-4974

L405-417 – Can you explain better why Dyngjujökull shows positive albedo trend? Is it

after the Holuhraun eruption and reduction of dust events from Dyngjusandur towards the glacier?

L419-426 – Figure 11 and discussion. Doesn't this show that warm, dry and dusty year as 2019 have similar impacts on albedo as volcanic eruption years?

L427 – Conclusions – It would be beneficial to conclude in one sentence also the difference in influence of tephra after eruption and dust during dusty year as 2019 on albedo.

L472- References should be ordered in ascending order (Palsson et al., Schmidt et al., Stroeve et al, need to be corrected).

L549-551 – Liu is not relevant reference in the text. They do not refer to volcanic ash from eruption, but general dust event. Consider to remove this from the reference list.

L554 – remove 'a' in Matlab, 2017a

L566 – Thorsteinsson et al., 2017 should be under T in the reference list, not under P.

---

## Author Comment (AC1) · 23 Jun 2020

Referee 2 (RC2):  Pavla Dagsson Waldhauserova, pavla@lbhi.is

Author response: 12.06.2020

The authors provide well-written detailed study on albedo changes of all Icelandic major glaciers using a comparison of MODIS snow albedo products and in situ measurements. This study could also serve as a comprehensive review of rapidly changing glaciers in Iceland with focus on impacts on their changing albedo. It brings insights into albedo analysis in problematic cloud-obscured region while providing novel findings on linear albedo gradients and dust plume shape patterns on snow and ice. Direct impacts of explosive volcanic eruptions as well as severe and moderate dust storms on the glaciers are evaluated. Additionally, possible indirect impacts of effusive eruptions such as Holuhraun 2014-2015 are suggested. It is clear that the authors know perfectly the local environment and its past. The data from the MODIS products were carefully screened during extensive manual quality control and results were evaluated with valuable data from in situ ICE-GAWS network. The greatest contribution of this study is that the data set does not only include major explosive eruptions and cold years, but it includes extremely rare year 2019, dry and dusty in the southern part of Iceland. This allows the authors to compare the impacts of volcanic ash and general volcanic/glacial dust on the albedo at the same level. There are minor errors in references that should be stated in ascending order and several references could be added. I would recommend publication after minor revisions.

Author response:

First, we would like to thank reviewer 2 (RC2) for very useful comments and a general positive feedback about our submitted manuscript. Your summary of the paper matches very well with our intended scope and deliverables.

**Specific comments:**

**L18-95 – Introduction**

Consider to add studies on snow albedo reductions due to volcanic dust, eg. Meinander et al., 2014, Peltoniemi et al., 2015, Dagsson-Waldhauserova et al., 2015, Zubko et al., 2019).

Kylling et al., 2018 calculated the instantaneous radiative forcing of the bottom of the atmosphere due to mineral dust deposited on snow as 0.135 W m-2.

Kylling A., Groot Zwaaftink, C. D., Stohl, A., 2018. Mineral dust instantaneous radiative forcing in the Arctic. Geophysical Research Letters, 45. doi: 10.1029/2018GL077346.

Peltoniemi, J. I., Gritsevich, M., Hakala, T., Dagsson-Waldhauserová, P., Arnalds, Ó., Anttila, K., Hannula, H.-R., Kivekäs, N., Lihavainen, H., Meinander, O., Svensson, J., Virkkula, A., de Leeuw, G., 2015. Soot on snow experiment: bidirectional reflectance factor measurements of contaminated snow. The Cryosphere 9, 3075-3111.
Dagsson-Waldhauserova, P., Arnalds, O., Olafsson, H., Hladil, J., Skala, R., Navratil, T., Chadimova, L., Meinander, O., 2015. Snow-dust storm A case study from Iceland, March 7th 2013. Aeolian Research 16, 69–74.

Meinander, O., Kontu, A., Virkkula, A., Arola, A., Backman, L., DagssonWaldhauserová, P., Järvinen, O., Manninen, T., Svensson, J., de Leeuw, G., and LepC2 päranta, M., 2014. Brief Communication: Light-absorbing impurities can reduce the density of melting snow. The Cryosphere 8, 991-995.

Zubko, N., Muñoz, O., Zubko, E., Gritsevich, M., Escobar-Cerezo, J., and Berg, J., 2019. Light scattering from volcanic-sand particles in deposited and aerosol form. Atmos. Env. 215, 116813. doi: 10.1016/j.atmosenv.2019.06.051

Author response:
We will incorporate these suggested references into the introduction part of the paper as they are highly relevant and will improve the manuscript.

L40 and L85 remove 'a' in Wittmann et al., 2017a. Consider to add Gascoin at al., 2017 here.

Author response:
References will be updated accordingly

L40-41 – 'surface albedo IS the dominating factors' - change ARE->IS, FACTORS- >FACTOR

Author response:
Will be changed accordingly

L44-45 – ..but it IS limited. . .

Author response:
Will be updated accordingly

L47 – Stroeve et al. 2001? As in reference, not 2002.

Author response:
References will be updated accordingly

L70-72 – Can you please rephrase the sentence or cut into two sentences. It is difficult to understand.

Author response:

Sentence is:
*To confirm this hypothesis, in-situ data and higher resolution data from Landsat 5 Thematic Mapper (TM) sensor were compared as well showing greater variability in surface albedo implying that the scale of the albedo variations is larger than the AVHHR pixel (1.1 km) could resolve.*

Rewrite:
*To confirm this hypothesis, Reijmer et al. (1999) compared in-situ data and higher spatial resolution remote sensing data from Landsat 5 Thematic Mapper (TM) sensor. The result showed greater variability in surface albedo implying that the scale of the albedo variations is larger than the AVHHR pixel (1.1 km) could resolve.*

L144 – Van Den Broeke et al., 2004 a,b?

Author response:
References will be updated accordingly

L164 – Table 1 – What do you mean by 'average location'?

Author response:
Annually when the GAWS stations are installed in the field they are not in the exact same location from on year to another. This can vary between a few tens to hundred meters. Stations can also move during the melt season due to ice flow. We calculate the average locations, mean value of these locations for pixel data extraction instead of posting annual values.

We will add a sentence in the caption of the table explained the meaning.

L192-193 – opening brackets are missing

**Author response:**
Will be added

L229-230 – Do you mean annual melt rates here?

**Author response:**
…high melt rates... refers to summer melt rates indicating that large elevation changes can be expected during summer resulting in tilting of the instruments.

L230 – Sand particles have certain size resolution, maybe 'dust' is better here. Or 'sand and dust'.

**Author response:**
Sentence is:
*Large sand and tephra covered areas have been observed…*

Suggested rewrite:
*Large sand, dust and tephra covered areas have been observed…*

L253– Small scale spatial variability of albedo could be also discussed here. See Hartl et al., 2020. Hartl, L., Felbauer, L., Schwaizer, G., and Fischer, A.: Small scale spatial variability of bare-ice albedo at Jamtalferner, Austria, The Cryosphere Discuss., https://doi.org/10.5194/tc-2020-92, in review, 2020.

**Author response:**
Will be added as a reference

L289-308 – Linear albedo gradients are really important and well discussed here. However, the role of local impurities should be also mentioned here. General lower albedo values at certain parts of Hofsjökull, Langjökull and Myrdalsjökull coincides well with location of dust source areas described in Arnalds et al., 2016, and classified as severe or extremely severe erosion areas. This should be also included here in the discussion. There is also work from Antarctica showing the vertical gradient of local dust impurities on glacier that could be discussed here. See Kavan et al., 2020.

Kavan, J., Nyvlt, D., Láska, K., Engel, Z., and Knazková, M. (2020) High latitude dust deposition in snow on the glaciers of James Ross Island, Antarctica. Earth Surf. Process. Landforms, https://doi.org/10.1002/esp.4831

**Author response:**
Correct, adding the following sentence:

*Local lower albedo gradients at Hofsjökull (SE), Langjökull (S) and Mýrdalsjökull (S) coincide well with documented locations of severe or extremely severe dust source areas described in Arnalds et al., 2016.*

L319-321 – General trends in annual albedo (lowest values vs. highest values) correspond to the long-term dust storm frequency studies in Iceland. For evaluation, consider these three studies:

Nakashima, M. and Dagsson-Waldhauserová, P., 2019. A 60 Year Examination of Dust Day Activity and Its Contributing Factors From Ten Icelandic Weather Stations From 1950 to 2009. Frontiers in Earth Science 6, 245-252. DOI:10.3389/feart.2018.00245

Butwin, M.K., von Löwis, S., Pfeffer, M., and Thorsteinsson, Th., 2019. The Effects of Volcanic Eruptions on the Frequency of Particulate Matter Suspension Events in Iceland. Journal of Aerosol Science 128, 99-113.

Dagsson-Waldhauserova, P., Arnalds, O., Olafsson, H., 2014. Long-term variability of dust events in Iceland. Atmospheric Chemistry and Physics 14, 13411-13422. DOI:10.5194/acp-14-13411-2014.

Author response:
For the cluster of glaciers L319-321 refers to (South coast glaciers) there is no significant trend in annual values, especially if the influence of the volcanic eruptions in 2010, 2011 and the residual effect in 2012 is removed.

However we will add long-term dust storm frequency studies in Iceland as a discussion point in Chapter 3.4 Trends of albedo

L322-323 – Such unstable erosive surfaces are defined as 'dust hot spots' and it has been shown that dust events occur frequently in southern parts of Iceland in winter. Examples here:

Dagsson-Waldhauserova, P., Arnalds, O., Olafsson, H., 2014. Long-term variability of dust events in Iceland. Atmospheric Chemistry and Physics 14, 13411-13422. DOI:10.5194/acp-14-13411-2014

Dagsson-Waldhauserova, P., Renard, J.-B., Olafsson, H., Vignelles, D., Berthet, G., Verdier, N., Duverger, V., 2019. Vertical distribution of aerosols in dust storms during the Arctic winter. Scientific Reports 6, 1-11.

Dagsson-Waldhauserova, P., Arnalds, O., Olafsson, H., Hladil, J., Skala, R., Navratil, T., Chadimova, L., Meinander, O., 2015. Snow-dust storm A case study from Iceland, March 7th 2013. Aeolian Research 16, 69–74.

Author response:
We will add these as references to further support our discussion in L322-323

L329 – delete 'r' in severer

Author response:
Yes

L329-332 – Just to comment. There are few cases when Drangajökull and Westfjords receive dust from the dust hot spots in central and South Iceland. Such events were captured by satellite or by dust model frequently in 2019.

Author response:
Figure 7 and 8 reflects this showing lower annual albedo values for 2019 for Drangajökull. In general, we would still consider Drangajökull to be "closest" of the Icelandic glaciers to have albedo driven by snow metamorphism even though dust events can take place. We have also add a sentence (See RC2 comment on L427 – Conclusions ) that highlights this.

L353 – '>30%' Did you mean < 30%

Author response:
Yes

L375 – Liu et al. (2014) do not really refer to volcanic ash from eruption, but dust event with maybe some relicts of ash. Their sample was collected in 2013 and they describe a dust event in 2013. I would suggest removing this from the references here.

Author response:
Yes, agreed

L380-381 – It was also induced by high dust storm activity in that area, see Möller et al., 2019. Volcanic ash is usually being removed fast from surfaces in Iceland, in < 1 year. See Butwin et al., 2019 or Arnalds et al., 2013.

Butwin, M.K., von Löwis, S., Pfeffer, M., and Thorsteinsson, Th., 2019. The Effects of Volcanic Eruptions on the Frequency of Particulate Matter Suspension Events in Iceland. Journal of Aerosol Science 128, 99-113.

Arnalds, O., Thorarinsdottir, E.F., Thorsson, J., Dagsson-Waldhauserova, P., Agustsdottir, A.M., 2013. An extreme wind erosion event of the fresh Eyjafjallajokull 2010 volcanic ash. Nature Scientific Reports 3, 1257.

Author response:

Sentence is:
No eruption occurred in 2012 but residual effects were observed as ash deposits from previous eruptions were carried with the prevailing wind directions, enhancing melt due to the lowering of albedo.

Rewrite:

No eruption occurred in 2012 but residual effects were observed as ash deposits from previous eruptions were carried with the prevailing wind directions and high dust storm activity reported in the area, enhancing melt due to the lowering of albedo (Butwin er al. 2019, Möller et al., 2019).

L382 – Wittmann et al. (2017a). Why 'a'?

Author response:
Typographical error, will be fixed

Figure 8 – Correct the title – delete 'for the'?

Author response:
Typographical error, will be fixed

L400 – 1999 Hekla – Are you talking about 26th Feb 2000 Hekla eruption here?

Author response:
Yes, 1999 should be 2000, will be fixed

L386-398 – When discussing dust influence on the albedos, you can also include that not only volcanic ash can be lifted to high altitudes and transported long distances. It is also Icelandic volcanic dust that can reach several km heights and travel long distances of thousands of km:

Dagsson-Waldhauserova, P., Renard, J.-B., Olafsson, H., Vignelles, D., Berthet, G., Verdier, N., Duverger, V., 2019. Vertical distribution of aerosols in dust storms during the Arctic winter. Scientific Reports 6, 1-11.

Djordjevic D., Toši ´ c I., Sakan S., Petrovi ´ c S., Ä ´ Ruri ˇ ci ˇ c-Milankovi ´ c J., Finger D.C. and ´ Dagsson-Waldhauserová P. 2019. Can Volcanic Dust Suspended From Surface Soil and Deserts of Iceland Be Transferred to Central Balkan Similarly to African Dust (Sahara)? Frontiers in Earth Sciences 7, 142-154.

Moroni B., Ólafur Arnalds, Pavla Dagsson Waldhauserová, Crocchianti, S., Vivani R., and Cappelletti, D. 2018. Mineralogical and chemical records of Icelandic dust sources upon Ny-Ålesund (Svalbard Islands). Frontiers in Earth Science 6, 187-219.

Beckett, F., Kylling, A., SigurÃˇrardóttir, G., von Löwis, S., and Witham, C., 2017. Quantifying the mass loading of particles in an ash cloud remobilized from tephra deposits on Iceland, Atmos. Chem. Phys., 17, 4401-4418.

Ovadnevaite J., Ceburnis D., Plauskaite-Sukiene K., Modini R., Dupuy R., Rimselyte I., Ramonet R., Kvietkus K., Ristovski Z., Berresheim H., O'Dowd C.D., 2009. Volcanic sulphate and arctic dust plumes over the North Atlantic Ocean. Atmospheric Environment 43, 4968-4974

Author response:
We will incorporate this to the manuscript. *Volcanic ash and dust…*

L405-417 – Can you explain better why Dyngjujökull shows positive albedo trend? Is it after the Holuhraun eruption and reduction of dust events from Dyngjusandur towards the glacier?

Author response:
Various influencing factors could contribute to a positive Dyngjujökull albedo trend. One of those possibly the changes due to the Holuhraun eruption as mentioned in L392-396 in the manuscript:

*"In 2014–15, the lava flow field of the Holuhraun non-explosive eruption covered about 84 km2 of volcaniclastic sandy desert and proglacial areas north of Vatnajökull. Since then, similar plume shaped albedo anomalies were not observed in the data. It is probable that the extent of the lava flow field reduces the dust production of this area significantly, although this cannot be quantified at this point in time, more data over a range of climatologies are needed to fully understand the impact of the Holuhraun eruption on dust production"*

But it could also be related to other climatological variables such as the variability in winter precipitation, melt onset and snowfall during late summer. The figure below shows the most recent mass balance values for Dyngjujökull among others. Since 2004-05 net mass balance has a mild upwards trend although non-significant constrained by high summer melt years for 2004 (Gjálp), 2010 (Eyjafjallajökull) and 2012 and low melt in 2015.

This is a very interesting question to discriminate the actual influencing factors driving this trend but we feel it needs a more detailed investigation than the scope of the study to be able to state anything definite.

[Figure]

Figure 16. Specific mass balance record for Vatnajökull outlets 1991_92-2018_19.

L419-426 – Figure 11 and discussion. Doesn't this show that warm, dry and dusty year as 2019 have similar impacts on albedo as volcanic eruption years?

Author response:
Depending on the timing of the deposits and extent this is true. In the cases where the active melting area (ablation zone) is extended significantly as happens during eruptions, more melt is expected. A key factor in the 2019 melt enhancement is how early seasonal snow is melted exposing these erosive surfaces.

L427 – Conclusions – It would be beneficial to conclude in one sentence also the difference in influence of tephra after eruption and dust during dusty year as 2019 on albedo.

Author response:
L436 says:
*Icelandic glacier albedo was observed to be influenced by variability in climate, tephra deposits from volcanic eruptions, and airborne dust from widespread unstable sandy surfaces which are subject to frequent wind erosion and dust production.*
We suggest adding the following to L426 with the discussion about Figure 11:

*Extensive dust transport to the glacier surface, as seen in the melt season of 2019, had similar overall albedo lowering effect as during the eruption years in 2010 and 2011 for Vatnajökull, Langjökull, Eyjafjallajökull and Drangajökull specifically. It is though noted that during volcanic eruptions albedo lowering is generally more localized while extensive dust transport tends to be more global.*

L472- References should be ordered in ascending order (Palsson et al., Schmidt et al., Stroeve et al, need to be corrected).

Author response:
Yes

L549-551 – Liu is not relevant reference in the text. They do not refer to volcanic ash from eruption, but general dust event. Consider to remove this from the reference list.

Author response:
We will correct this.

L554 – remove 'a' in Matlab, 2017a

Author response:
We will correct this.

L566 – Thorsteinsson et al., 2017 should be under T in the reference list, not under P

Author response:
We will correct this.

---

## Author Comment (AC2) · 23 Jun 2020

Referee 1 (RC1): Simon Gascoin, simon.gascoin@cesbio.cnes.fr

Author response: 12.06.2020

This paper presents the application of MODIS snow albedo products to characterize the spatial-temporal variability and trends of glacier albedo in Iceland. The albedo data are derived from the M*D10A1 products after interpolating missing data due to the frequent cloud cover. The topic is interesting since Icelandic glaciers are frequently exposed to volcanic ashes deposition causing large albedo changes and thereby modulating their response to climate forcing (Schmidt et al. 2017). A strength of the study is the extensive in situ dataset that was used to evaluate the MODIS products (20 AWS).

My main concern regarding this study is the apparent lack of novelty with respect to previous works by Möller et al. (2014) and Gascoin et al. (2017) who also studied the albedo changes over Icelandic glaciers. Some figures in the manuscript provide similar information as Gascoin et al. (2017).

In particular the introduction does not clearly state why it was needed to go beyond previous studies by Möller et al. (2014) and Gascoin et al. (2017). I see some differences that could indeed justify this new study.

The authors used M*D10A1, while the latter studies used MCD43A3. However, the authors should strengthen this part of the manuscript by providing a detailed comparison of both products. As it stands, the results cannot be compared to those reported by Gascoin et al. (2017) mainly because the authors computed the RMSE and correlations at the monthly time step whereas we used daily values (see L245 "The comparison presented here is in fact similar to previous work on Icelandic glaciers by Gascoin et al. (2017) where the MCD43A3 was evaluated with RMS errors ranging from 8–21%.").

First, we would like to thank reviewer 1 (RC1) for useful comments and feedback about our submitted manuscript.

The original scope of the work was to quantify and assess the influence of volcanic activity and dust deposits on the surface mass balance of Icelandic glaciers using MC43A3 remotely sensed albedo, following in many aspects the work done by Gascoin et al. (2017). During this work, limitations of MCD43A3 for Icelandic glaciers where exposed. Below we hope to address those limitations.

Schmidt (et al. 2017) highlighted the importance of accurate glacier albedo for estimates of surface mass balance for Icelandic glaciers and similarly, this has also been observed in various hydrological models' efforts by the National Power Company in Iceland for many years.

Work done by Schmidt et al. 2017 used average values of MCD43A3 albedo as a background information for bare ice albedo to further improve the lowest albedo expected per pixel. It does not attempt to model the impact of dust deposit events, neither originated from exposed erosive surfaces in the proglacial areas nor the influence of volcanic eruptions while this is what we strive to do eventually.

As detailed in Gascoin et al. (2017) and Gunnarsson et al. (2019) a major challenge for remote sensing in Iceland is cloud cover, even though data from both Aqua and Terra are used cloud cover/no data pixels is still high. In addition to this, the strict processing criteria of the multi-look product from MCD43 reduces usable pixels even further, especially at higher elevations for Vatnajökull. An example of this is clearly shown in Figure 1 (below) where the pixel density for the melt season in 2019 (MJJA) is shown, for the combination of pixels available from MOD10A1 and MYD10A1. In comparison, Figure 2 (below) shows the pixel density for the melt season in 2019 (MJJA) for the MCD43A3 product. Essentially, this is the main reason we developed a new processing pipeline utilizing all data that is available, but also allowing for more tailored methodology to filter and reject pixels which is limited for MCD43.

It is hard to see how the indicated lack of novelty in our manuscript relates to the work done by Möller et al. (2014). Möller et al. (2014) investigate a single event (2004 Grímsvötn eruption) compared to an ash-dispersal dataset obtained from in situ measurements on the ice cap to develop a empirically based modelling approach to describe the albedo decrease across the glacier surface caused by the deposited tephra. The work done by Möller et al. (2014) is cited in our submitted manuscript.

Note must also be taken that MCD43A3 and MOD10A1 albedos are differently processed although obtained from the same sensor/daily surface reflectance product.

The scope of the paper is not to compare M*D10A1 or MCD11 to MCD43 albedo. It is aimed to develop a method to provide gap filled spatial-temporal continuous products to model near real time surface ablation and runoff in glacier fed rivers by direct albedo assimilation. That is the novelty of the paper.

A few key points highlighting the difference from our submitted manuscript and Gascoin et al. (2017).

- Melt increase from dust/ash deposit events are mostly observed to extend the active melt area of the glaciers, due to light absorbing impurities deposited in the accumulation area. This is why it is also important to represent albedo in the accumulation area better than MCD43 is able to.

- We have an albedo product that has a 2-5 day lag compared to the 14-16 day lag by MCD43A3. This is probably as close to "real time" as possible allowing usage of the albedo data in operational context.

- A much more detailed study is provided in our manuscript analyzing patterns and spatial trends of albedo than provided in Gascoin et al. (2017). Relations to elevation, monthly statistics and trends over time as well as temporal properties are reported. Individual dust events from documented erosive surfaces are identified and speculations relating the influence of the newly Holuhraun lava flow in 2015 are set forth among various other details.

- Seven year record for comparison and analysis for Icelandic glaciers are added including the cold summer of 2015, resulting in the first positive mass balance in 20 years at the time, and also the extreme dust deposit summer of 2019, is a good addition to the range of data in Gascoin et al. (2017) which spans 2000 - 2012.

We realize that MCD11 data is not perfect and there is a reason for the strict filtering criteria in MCD43. While MCD43 allows very limited improvement as it is a ready-made product the processing pipeline for MCD11 allows for more detailed filtering and rejection of cloud misclassified pixels. This methodology also allows for future improvements in the filtering criteria.

[Figure]

Figure 1 – MOD10A1 and MYD10A1 available pixels for MJJA 2019 for the largest Icelandic glaciers

[Figure]

Figure 2 - MCD43 available pixels for MJJA 2019 for the largest Icelandic glaciers.

[Figure]

Figure 3 – Snapshot of calculated albedo and MCD43A3 MODIS albedo for a sub catchment at Brúarjökull glacier in Vatnajökull.

Looking at Tab B1 it seems that M*D10 products are more accurate than MCD43?

Author response:
There are some cases where the $R^2$ is better for MOD10A1 as well as lower RMSE values. Overall though, MCD43A3 albedo has better or equal R2 performance, in 15 out of 20 sites compared.

Also, an important aspect is that MCD43 provides albedo over all land masses, whereas M*D10A1 provides only albedo of the pixels that are detected as snow covered. This can be an issue in Iceland where large regions of glaciers may not be detected as snow due to the tephra layer. This issue should be investigated to make sure that the MCD11 product is not interpolating the albedo of clear-sky, snow-free pixels.

Author response:
Yes, correct, this is a very good comment. This has been visually investigated near the eruption sites at Grímsvötn for 2004, 2011 and Eyjafjallajökull eruption in 2010. In general, the random forest model is capable to estimate reasonable values for the thick tephra covered areas when they are not detected as snow, especially near the eruption site in 2011 in Vatnajökull.

To ensure less misclassification from clouds or tephra plumes during the eruption in these areas the local outlier thresholds applied are relieved allowing more range of expected values, especially lower values at higher elevations.

One weakness in our method is that during an eruption it might be hard to know the active extent of a tephra fallout that provides isolation to the surface. In a similar way tephra plumes discharged into the atmosphere with high tephra concentrations might further induce misclassifications. This is partially solved by Möller et al. (2014) fusing MOD10A1 and MCD43A3 albedos which might be a better future solution during eruptions and production of large thick tephra covered areas.

To highlight these problems, we will add the following sentence in L182:

*In areas near the eruption sites in 2010 and 2011 the local outlier thresholds applied are adjusted allowing more range of expected values, especially lower values, to include the effect for tephra deposits to the glacier surface.*

The trends should be masked or marked where MK test is not significant (Figure 10).

Author response:
We will add this to Figure 10

The improvements in MCD11 albedo with respect to the original product are very small (about 0.01 RMSE, Tab.2). In addition, it is indicated (L181) that the thresholds for outliers rejection were manually adjusted so the conclusions remain limited to this study. The main benefit of MCD11 is rather that it is a gap-free product which facilitates the utilization of the data.

Author response:
This is true and the main scope of the work. This is similar to Box et al. 2012 where local outlier thresholds are applied to improve the albedo retrieval for Greenland. We aim at improving the albedo retrievals for Icelandic glaciers, not glaciers worldwide.

The authors indicate that a motivation of their work is the integration of this albedo product in operational snow melt runoff model. It would be useful to have more background information on this aspect. What albedo is currently used by Landsvirkjun or other agencies? Is the developed product compliant with operational context if there is a lag of at least 5 days before updating the albedo (since temporal interpolation is based on a 10 days window)?

Author response:
Currently, albedo is calculated by a recent physically based broadband albedo parameterization (Gardner and Sharp, 2010). It is dependent on the five variables; specific surface area of snow (SSA), concentration profile of light absorbing carbon (or equivalent dirt) within the snow pack, cloud optical thickness, solar zenith angle and snow depth.

An example of calculated albedo results is provided in Figure 3 where MCD43A3 albedo is compared to glaciated sub-catchment on Brúarjökull. The room for improvement is very visible.

Availability for near real-time data is also important in operational context. MCD43 generally has a longer lag time, 12-14 days while M*D10A1 data is available with a 1-2-day lag. In the manuscript data is processed from a center date using data from 5 days into the past and future resulting in a 5-day lag from the current day. Currently the MCD11 product runs operationally daily with a 2-day lag. To do that, a modification of the process pipeline uses all available data 11 days back in time bridging from the conventional MCD11 to MCD11OPER which is then

overwritten when sufficient data is available to process with the pipeline as outlined in the manuscript. This is not a perfect solution but aims at having near real time estimations of albedo and albedo changes. Especially in the case of volcanic eruptions response times can be reduced to model the possibilities of floods due to melt enhancement and operational strategies for reservoir operation.

**Minor comments**
L31 in an maritime climate

Author response:
Will be fixed

L34: Seasonal glacier melt : what does it mean: seasonal snow and ice melt from the glacier area

Author response:
Yes, it is the amount the glacier melts seasonally.

L41: are
Author response:

Sentence is:
For Icelandic glaciers, surface albedo are the dominating factors governing surface melt annual variability…

Rewrite:
For Icelandic glaciers, surface albedo is the dominating factor governing surface melt annual variability…

L93: this paragraph gives me the impression to come out of the blue. The objective should be more clearly linked to the literature review and identified knowledge gaps.

Author response:
This paragraph summarizes the main objectives of the study based on the introduction for the convenience of the reader.

L153: "Daily averages" is not the correct wording if it refers to of hourly albedo values. I understand from the above paragraph that the daily albedo was in fact calculated from daily sums of incoming and reflected radiation (which is recommended to reduce measurement noise).

Author response:

For validation and comparison in the manuscript we calculated as the running 24-hour sum of upward shortwave divided by the running 24-hour sum of the downward shortwave as detailed in L135-137.

We will remove the following sentence from L153 as is originates from a version of the paper where we had modelling results to not create confusion:

*Daily averages were calculated from hourly averages if at least 20 hourly values were available and monthly averages were calculated from daily averages if 24 values or more were available.*

L168: what is a "median based statistical rejection of outliers."

Author response:
L177 explains better what median based statistical rejection of outliers does. Essentially this is to remove noise from the stacked pixels. Points that are larger or smaller than the median value of a given pixel stack are removed as outliers.

L173: I don't think you need these references to justify this general statement.

Author response:
References will be removed

L184: these pixels are not unclassified, since they are classified as cloud.

Author response:
Correct, *unclassified pixels* will be changed to *pixels classified as clouds*

L185: this approach is very similar to our algorithm for cloud pixels interpolation in MOD10 products (Gascoin et al. 2015). We used the same predictors. It should be cited if it has inspired your own algorithm.

Author response:
These are quite common predictors used in various studies we have researched and cited in the study. Indeed Gascoin et al. 2015 uses similar methodology.

L188 Correspondingly reads a bit odd here

Author response:
Correspondingly will be changed

L191 "monthly, basis"

Author response:
We will move the comma

L204: The calculations were

Author response:
Were will be changed to are

L215-220 the whole paragraph should be removed (it is method, not results)

Author response:
We will remove the paragraph

L246: results are not directly comparable (daily vs. monthly) (see my main comments)

Author response:

Sentence is:
The comparison presented here is in fact similar to previous work on Icelandic glaciers by Gascoin et al. (2017) where the MCD43A3 was evaluated with RMS errors ranging from 8–21%.

Rewrite:
The comparison presented here is in fact similar to previous work on Icelandic glaciers by Gascoin et al. (2017) where the MCD43A3 was evaluated with RMS errors ranging from 8–21%, although the results from Gascoin et al. (2017) are based on daily values.

L253: "indicating high sub-pixel albedo variability" This is a bit vague and unexpected comment since large areas of Icelandic ice caps have a rather homogeneous surface (in comparison with Alpine glaciers for example). We studied albedo subpixel variability from Landsat data to explain the discrepancy between AWS measurements and MODIS retrieval.

Author response:
Yes, more details are needed here. We will make the following change:

Sentence is:
*Sub-pixel variability has been investigated by Reijmer et al. (1999) and Gascoin et al. (2017) for the Icelandic glaciers indicating high sub-pixel albedo variability.*

Rewrite:
*Sub-pixel variability has been investigated by Reijmer et al. (1999), Pope et al. (2016), and Gascoin et al. (2017) for the Icelandic glaciers. Results indicate higher sub-pixel albedo variability in the bare-ice areas, especially where stratified dirt bands and debris is observed while less variability is reported in the flat surroundings at higher elevations.*

L273 experienced as an smoothing

Author response:
Fixed

Fig 3: a similar figure can be found in Gascoin et al 2017

Author response:
These figures show similar patterns. We suggest keeping this figure in the manuscript as it illustrates the cloud cover during the active melt season (MJJA) in Iceland not the whole data period (Feb to Nov).

The figure in Gascoin et al 2017 shows data availability for the whole year including the period during polar darkness when no data are available providing different information related to cloud cover. It also does not detail the cloud cover over the bare ice areas that form as the winter snow is melted from the dirty ice-covered surface and its development into the melt period.

Fig 4, 6, 7: rainbow colormaps are not recommended (see e.g. https://www.nature.com/articles/519291d)

Author response:
This is a good point and we spent a considerable amount of time selecting colormaps. Our conclusion was to highlight patterns in the data, bare ice areas, estimations of ELA and accumulation areas, a rainbow colormap is the best way to do so. This follows the examples by Box et al. 2012, Stroeve at al. 2013 and Riihelä et al. 2019.

Fig 6: the figure does not display correctly on my computer, I suggest to replace it by a bitmap (raster) version

Author response:
A final manuscript will have all figures as a raster

L440: this sentence should be removed or reformulated since there is no information on glacier mass balance in this study
Author response:
Correct, this will be removed as data regarding mass balance has been removed.

L462: Do you mean when MODIS will stop operating? Note that the successor of MODIS is rather VIIRS.
Author response:

We realize that VIIRS has operational data but look also towards using data from the SICE project (http://snow.geus.dk/) to take full potential of the twice per day overpass over Iceland.

We suggest modifying the sentence to:
... such as Sentinel 3 and VIIRS, to extend...

---

## Author Response (AR1)

Referee 1 (RC1): Simon Gascoin, simon.gascoin@cesbio.cnes.fr

Author response: 12.06.2020
Updated author response: 28.09.2020

This paper presents the application of MODIS snow albedo products to characterize the spatial-temporal variability and trends of glacier albedo in Iceland. The albedo data are derived from the M*D10A1 products after interpolating missing data due to the frequent cloud cover. The topic is interesting since Icelandic glaciers are frequently exposed to volcanic ashes deposition causing large albedo changes and thereby modulating their response to climate forcing (Schmidt et al. 2017). A strength of the study is the extensive in situ dataset that was used to evaluate the MODIS products (20 AWS).

My main concern regarding this study is the apparent lack of novelty with respect to previous works by Möller et al. (2014) and Gascoin et al. (2017) who also studied the albedo changes over Icelandic glaciers. Some figures in the manuscript provide similar information as Gascoin et al. (2017).

In particular the introduction does not clearly state why it was needed to go beyond previous studies by Möller et al. (2014) and Gascoin et al. (2017). I see some differences that could indeed justify this new study.

The authors used M*D10A1, while the latter studies used MCD43A3. However, the authors should strengthen this part of the manuscript by providing a detailed comparison of both products. As it stands, the results cannot be compared to those reported by Gascoin et al. (2017) mainly because the authors computed the RMSE and correlations at the monthly time step whereas we used daily values (see L245 "The comparison presented here is in fact similar to previous work on Icelandic glaciers by Gascoin et al. (2017) where the MCD43A3 was evaluated with RMS errors ranging from 8–21%.").

Author response:

First, we would like to thank reviewer 1 (RC1) for useful comments and feedback about our submitted manuscript.

The original scope of the work was to quantify and assess the influence of volcanic activity and dust deposits on surface mass balance of Icelandic glaciers using MC43A3 remotely sensed albedo, following in many aspects the work done by Gascoin et al. (2017). Also, in an operational context for hydropower generation, MCD43A3 has been used in relative comparison to qualitatively assess near future (days/weeks) melt potential, similar to data for Greenland by DMI, DTU and GEUS (http://polarportal.dk/en/greenland/surface-conditions/). Note that Polar Portal uses MOD10A1. During this work, limitations of MCD43A3 for Icelandic glaciers where exposed. Below we hope to address those limitations.

Schmidt et al. 2017 has highlighted the importance of accurate glacier albedo for estimates of surface mass balance for Icelandic glaciers and similarly, these challenges have also been observed in various hydrological models' efforts by the National Power Company in Iceland for many years.

Work done by Schmidt et al. 2017 used average values of MCD43A3 albedo as a background information about bare ice albedo to further improve the lowest albedo expected per pixel. It does not include efforts to model the impact of dust deposit events, either originated from exposed erosive surfaces in the proglacial areas nor the influence of volcanic eruptions while this is what we strive to do eventually.

As detailed in Gascoin et al. (2017) and Gunnarsson et al. (2019) a major challenge for remote sensing in Iceland is cloud cover, even though data from both Aqua and Terra are used cloud cover/no data pixels is still high. In addition to this, the strict processing criteria of the multi-look product from MCD43 reduces usable pixel even further, especially at higher elevations for Vatnajökull. An example of this is clearly shown in Figure 1 (below) where the pixel density for the melt season in 2019 (MJJA) is shown, for the combination of pixels available from MOD10A1 and MYD10A1. In comparison, Figure 2 (below) shows the pixel density for the melt season in 2019 (MJJA) for the MCD43A3 product. Essentially, this is the main reason we developed a new processing pipeline utilizing all data that is available, but also allowing for more tailored methodology to filter and reject pixels which is limited for MCD43.

It is hard to see how the indicated lack of novelty in our manuscript relates to the work done by Möller et al. (2014). Möller et al. (2014) investigate a singular event (2004 Grímsvötn eruption) liked to an ash-dispersal dataset obtained from in situ measurements on the ice cap to develop a empirically based modelling approach to describe the albedo decrease across the glacier surface caused by the deposited tephra. The work done by Möller et al. (2014) is cited in our submitted manuscript.

Note must also be taken that MCD43A3 and MOD10A1 albedos are differently processed although obtained from the same sensor/daily surface reflectance product.

The scope of the paper is not to compare M*D10A1 or MCD11 to MCD43 albedo. It is to develop a method to provide gap filled spatial-temporal continuous products to model near real time influences of inflows to glacier fed rivers by direct albedo assimilation. That is the novelty of the paper.

A few key points highlighting the difference from our submitted manuscript and Gascoin et al. (2017).

- Melt increase from dust/ash deposit events are mostly observed to extent the active melt area of the glaciers, i.e. light absorbing impurities deposit in the accumulation area. This is why it is also important to represent the accumulation area better than MCD43 is able to.

- We have a product that has a 2-5 day lag compared to the 14-16 day lag by MCD43A3. This is probably as close to "real time" as we will get to be able to use the data in operational context.

- A much more detailed study is provided in our manuscript analyzing patterns and spatial trends of albedo than provided in Gascoin et al. (2017). Relations to elevation, monthly statistics and trends over time as well as temporal properties are reported. Individual dust events from documented erosive surfaces are identified and speculations relating the influence of the newly Holuhraun lava flow in 2015 are set forth among other various details.

- Seven years of comparison and analytics for Icelandic glaciers are added including the very cold summer of 2015 resulting in the first positive mass balance in 20 years at the time but as well the extreme dust deposit year in 2019 compared to the range of data in Gascoin et al. (2017) which spans 2000 - 2012.

We realize that MCD11 data is not perfect and there is a reason for the strict filtering criteria in MCD43. While MCD43 allows very limiting improvement as it is a ready-made product the processing pipeline for MCD11 allows for more detailed filtering and rejection of cloud misclassified pixels. This methodology also allows for future improvements in the filtering criteria.

[Figure]

Figure 1 – MOD10A1 and MYD10A1 available pixels for MJJA 2019 for the largest Icelandic glaciers

[Figure]

Figure 2 - MCD43 available pixels for MJJA 2019 for the largest Icelandic glaciers.

[Figure]

Figure 3 – Snapshot of calculated albedo and MCD43A3 MODIS albedo for a sub catchment at Brúarjökull glacier in Vatnajökull.

To accommodate the above comments from RC1 we have made the following changes to the manuscript:

L89:

*Cloud cover is a major challenge for remote sensing in Iceland, even though data from both Aqua and Terra are used, the amount of cloud-covered pixels is still high. For albedo derived from the MODIS MCD43A3 product, the strict processing criteria of the multi-look product reduce the number of usable pixels even further than collected by Aqua and Terra. This is especially true at higher elevations for Vatnajökull where persistent cloud cover is frequently observed, resulting in fewer valid albedo pixels during the melt season. Melt increase from dust and ash deposit events is observed to extend the active melt area of the glaciers, i.e. LAP deposit in the accumulation area, increasing melting. Therefore data from these areas are very important for monitoring and forecasting runoff from glaciers in Iceland. Lag times of MCD43A3 (14--16 days) make this less feasible for near-real-time monitoring and operational modelling, for example, in the case of a major dust deposit or volcanic eruption. Additionally, MCD43A3 is not gap-filled, requiring some post-processing prior to monitoring or hydrological modelling efforts*

Also in L100 when writing the studies objectives:

*This study aims to address some of the shortcomings of the MCD43A3 product for glaciers in Iceland and derive an albedo data set suitable for operational use as well as a scientific study of spatial and temporal variations in albedo. The daily M\*D10A1 products were chosen to increase temporal resolution, allowing for more flexibility in post-processing, statistical filtering and near-real-time data posting. There are two main objectives of the study. First, to create a gap-filled MODIS-based surface-albedo product for glaciers in Iceland for this time period from 2000 to 2019 validated with in-situ data suitable for the monitoring and modelling of glaciers in an operational context. Second, the resulting gap-filled product was used to analyse and quantify spatio-temporal patterns of albedo for Icelandic glaciers for the time period, with monthly statistics and a detailed interpretation of the variation of albedo with elevation and trends over time.*

And in the conclusion of the paper in L 463:

*Details are provided regarding spatial patterns and temporal trends, relations to elevation and monthly statistics adding to previous work by Gascoin et al. (2017) for 2000 to 2012.*

*In addition to the information provided in this document, we hope to have provided enough insights into these two different studies.*

Looking at Tab B1 it seems that M*D10 products are more accurate than MCD43?

**Author response:**
There are some cases where the $R^2$ is better for MOD10A1 as well as lower RMSE values. Overall though, MCD43A3 albedo has better or equal R2 performance, in 15 out of 20 sites compared. The outlier rejection of M*D10A1 data shows in many cases successful cloud rejection which is not rejected for MCD43A3.

Also, an important aspect is that MCD43 provides albedo over all land masses, whereas M*D10A1 provides only albedo of the pixels that are detected as snow covered. This can be an issue in Iceland where large regions of glaciers may not be detected as snow due to the tephra layer. This issue should be investigated to make sure that the MCD11 product is not interpolating the albedo of clear-sky, snow-free pixels.

**Author response:**
Yes, correct, this is a very good comment. This has been visually investigated near the eruption sites at Grímsvötn for 2004, 2011 and Eyjafjallajökull eruption in 2010. In general, the random forest model is capable to project reasonable values for the thick tephra covered areas when they are not detected as snow, especially near the eruption site in 2011 in Vatnajökull.

To ensure less misclassification from clouds or tephra plumes during the eruption in these areas the local outlier thresholds applied are relived allowing more range of expected values, especially lower values at higher elevations.

One weakness in our method is that during an eruption it might be hard to know the active extent of a tephra fallout that provides isolation to the surface. In a similar way tephra plumes discharged into the atmosphere with high tephra concentrations might further induce misclassifications. This is partially solved by Möller et al. (2014) fusing MOD10A1 and MCD43A3 albedos which might be a better future solution during these eruptions and production of large thick tephra covered areas.

To highlight these problems, we will add the following sentence in L194:

*During periods effected volcanic eruptions the outlier thresholds are not applied, allowing a greater range of expected values, especially lower values at higher elevations where tephra deposits were observed. In this study, this applies in melt seasons 2010 and 2011.*

The trends should be masked or marked where MK test is not significant (Figure 10).

**Author response:**
Figure 10 now shows masked areas where significant trends are observed

The improvements in MCD11 albedo with respect to the original product are very small (about 0.01 RMSE, Tab.2). In addition, it is indicated (L181) that the thresholds for outliers rejection were manually adjusted so the conclusions remain limited to this study. The main benefit of MCD11 is rather that it is a gap-free product which facilitates the utilization of the data.

**Author response:**
This is true and the main scope of the work. This is similar to Box et al. 2012 where local outlier thresholds are applied to improve the albedo retrieval for Greenland. We aim at improving the albedo retrievals for Icelandic glaciers.

The authors indicate that a motivation of their work is the integration of this albedo product in operational snow melt runoff model. It would be useful to have more background information on this aspect. What albedo is currently used by Landsvirkjun or other agencies? Is the developed product compliant with operational context if there is a lag of at least 5 days before updating the albedo (since temporal interpolation is based on a 10 days window)?

**Author response:**
Currently, albedo is calculated by a recent physically based broadband albedo parameterization (Gardner and Sharp, 2010). It is dependent on the five variables; specific surface area of snow (SSA), concentration profile of light absorbing carbon (or equivalent dirt) within the snow pack, cloud optical thickness, solar zenith angle and snow depth.

An example of calculated albedo results is provided in Figure 3 where MCD43A3 albedo is compared to glaciated sub-catchment on Brúarjökull. The room for improvement is very visible.

Availability for near real-time data is also important in operational context. MCD43 generally has a longer lag time, 12-14 days while M*D10A1 data is available with a 1-2-day lag. In the manuscript data is processed from a center date using data from 5 days into the past and future resulting in a 5-day lag from the artificial current day. Currently the MCD11 product runs operationally daily with a 2-day lag. To do that, a modification of the process pipeline uses all available data 11 days back in time bridging from the conventional MCD11 to MCD11OPER which is then overwritten when sufficient data is available to process with the pipeline as outlined in the manuscript. This is not a perfect solution but aims at having near real time estimations of albedo and albedo changes. Especially in the case of volcanic eruptions response times can be reduced to model the possibilities of floods due to melt enhancement and operational strategies for reservoir operation.

We have added discussion to this end in L479:
*The methodology allows for predictive and retrospective modes (Dozier et al. 2008), depending on the application. To use the albedo data for runoff forecasting for example, surface albedo*

*estimations using only data until the present (newest MODIS data) can be provided by applying the statistical filtering and gap-filling routines from today and backwards. Alternatively, in retrospective mode, best estimations can be provided for every day in a period.*

**Minor comments**
L31 in an maritime climate

Author response:
This paragraph has been remove to shorten the article a bit.

L34: Seasonal glacier melt : what does it mean: seasonal snow and ice melt from the glacier area

Author response:
Yes, it is the amount the glacier melts seasonally. This paragraph has been remove to shorten the article a bit.

L41: are
Author response:
Fixed

L93: this paragraph gives me the impression to come out of the blue. The objective should be more clearly linked to the literature review and identified knowledge gaps.

Author response:
This paragraph summarizes the main objectives of the study based on the introduction for the convenience of the reader. We have updated them with more details.

See L100:

This study aims to address some of the shortcomings of the MCD43A3 product for glaciers in Iceland and derive an albedo data set suitable for operational use as well as a scientific study of spatial and temporal variations in albedo. The daily M*D10A1 products were chosen to increase temporal resolution, allowing for more flexibility in post-processing, statistical filtering and near-real-time data posting. There are two main objectives of the study. First, to create a gap-filled MODIS-based surface-albedo product for glaciers in Iceland for this time period from 2000 to 2019 validated with in-situ data suitable for the monitoring and modelling of glaciers in an operational context. Second, the resulting gap-filled product was used to analyse and quantify spatio-temporal patterns of albedo for Icelandic glaciers for the time period, with monthly statistics and a detailed interpretation of the variation of albedo with elevation and trends over time.

L153: "Daily averages" is not the correct wording if it refers to of hourly albedo values. I understand from the above paragraph that the daily albedo was in fact calculated from daily sums of incoming and reflected radiation (which is recommended to reduce measurement noise).

**Author response:**

For validation and comparison in the manuscript we calculated as the running 24-hour sum of upward shortwave divided by the running 24-hour sum of the downward shortwave as detailed in L135-137.

We will remove the following sentence from L153 as is originates from a version of the paper where we had modelling results to not create confusion:

*Daily averages were calculated from hourly averages if at least 20 hourly values were available and monthly averages were calculated from daily averages if 24 values or more were available.*

L168: what is a "median based statistical rejection of outliers."

**Author response:**
L177 explains better what median based statistical rejection of outliers does. Essentially this is to remove noise from the stacked pixels. Points that are larger or smaller than the median value of a given pixel stack are removed as outliers.

We have merged and updated these sentences to: see L189

*On a pixel-by-pixel basis, the method Box et al. 2012 was applied to reject values exceeding 2 standard deviations from the 11 day temporally aggregated data stack. The method is only applied if 4 or more pixels in the data stack have valid albedo data. To prevent rejection of valid data, values within a certain threshold of the median were not rejected. The outlier thresholds were manually adjusted, mostly related to the elevation of the glaciers, ranging from 1 to 4\%, for higher to lower elevation, respectively. From the 22 potentially available values, the mean is calculated to represent the surface-albedo, after median-based statistical rejection of outliers. During periods effected volcanic eruptions the outlier thresholds are not applied, allowing a greater range of expected values, especially lower values at higher elevations where tephra deposits were observed. In this study, this applies in melt seasons 2010 and 2011.*

L173: I don't think you need these references to justify this general statement.

**Author response:**
References is removed

L184: these pixels are not unclassified, since they are classified as cloud.

**Author response:**
Correct, *unclassified pixels* will be changed to *pixels classified as clouds,* see L 198

L185: this approach is very similar to our algorithm for cloud pixels interpolation in MOD10 products (Gascoin et al. 2015). We used the same predictors. It should be cited if it has inspired your own algorithm.

**Author response:**
These are quite common predictors used in various studies we have researched and cited in the study. Indeed Gascoin et al. 2015 uses similar methodology.

L188 Correspondingly reads a bit odd here

**Author response:**
Correspondingly is changed to *Aspect was then calculated for each pixel* see L202

L191 "monthly, basis"

**Author response:**
Fixed

L204: The calculations were

**Author response:**
Fixed

L215-220 the whole paragraph should be removed (it is method, not results)

**Author response:**
Agreed, paragraph is removed

L246: results are not directly comparable (daily vs. monthly) (see my main comments)

**Author response:**

Sentence is:
The comparison presented here is in fact similar to previous work on Icelandic glaciers by Gascoin et al. (2017) where the MCD43A3 was evaluated with RMS errors ranging from 8–21%.

Rewrite: see L253
The comparison presented here is in fact similar to previous work on Icelandic glaciers by Gascoin et al. (2017) where the MCD43A3 was evaluated with RMS errors ranging from 8–21%, although the results from Gascoin et al. (2017) are based on daily values.

L253: "indicating high sub-pixel albedo variability" This is a bit vague and unexpected comment since large areas of Icelandic ice caps have a rather homogeneous surface (in comparison with Alpine glaciers for example). We studied albedo subpixel variability from Landsat data to explain the discrepancy between AWS measurements and MODIS retrieval.

**Author response:**
Yes, more details are needed here. We will make the following change:

Sentence is:
*Sub-pixel variability has been investigated by Reijmer et al. (1999) and Gascoin et al. (2017) for the Icelandic glaciers indicating high sub-pixel albedo variability.*

Rewrite: see L 262
*Sub-pixel variability has been investigated by Reijmer et al. (1999), Pope et al. (2016), and Gascoin et al. (2017) for Icelandic glaciers. The study by Reijmer et al. (1999), using AVHRR and Landsat TM data at Vatnajökull reported large systematic differences for some of the automatic weather stations on the ice, attributed to sub-pixel-scale variations in the albedo. Results implied that the scale of the albedo variations was smaller than the scale of the AVHRR and TM pixels. Pope et al. (2016) studied high-resolution (5 m) airborne multi-spectral data collected over Langjökull in 2007, with comparison to near-contemporaneous Landsat ETM+ and MODIS imagery showing albedo to be highly variable at small spatial scales. Work by Gascoin et al. (2017) suggested that the RMSE of the difference between the in-situ automatic weather station data and MODIS data tends to increase when the corresponding Landsat sub-pixel spatial variability is higher. Lower standard deviation values were consistently obtained where the surface was less heterogeneous (accumulation areas).*

L273 experienced as an smoothing

Author response:
Fixed

Fig 3: a similar figure can be found in Gascoin et al 2017

Author response:
These figures show similar patterns. We suggest keeping this figure in the manuscript as it illustrates the cloud cover during the active melt season (MJJA) in Iceland not the whole data period (Feb to Nov).

The figure in Gascoin et al 2017 shows data availability for the whole year including the period during polar darkness when no data are available providing different information related to cloud cover. It also does not detail the cloud cover over the bare ice areas that form as the winter snow is melted from the dirty ice-covered surface and its development into the melt period which we want to highlight.

Fig 4, 6, 7: rainbow colormaps are not recommended (see e.g. https://www.nature.com/articles/519291d)

Author response:
Relevant colormaps have been updated

Fig 6: the figure does not display correctly on my computer, I suggest to replace it by a bitmap (raster) version

Author response:
All figure are pdfs and eps now

L440: this sentence should be removed or reformulated since there is no information on glacier mass balance in this study
Author response:
Correct, this will be removed as data regarding mass balance has been removed.

L462: Do you mean when MODIS will stop operating? Note that the successor of MODIS is rather VIIRS.
Author response:
We realize that VIIRS has operational data but look also towards using data from the SICE project (http://snow.geus.dk/) to take full potential of the twice per day overpass over Iceland.

L485 has been update to:
*Finally, it is noted that the methodology applied in the study, based on MODIS data, can be applied to other satellite albedo products, such as VIIRS and Sentinel-3 as well as future missions, to extend the temporal range beyond the MODIS mission, allowing for short-term as well as long-term monitoring of albedo variations for glaciers in Iceland.*
The authors provide well-written detailed study on albedo changes of all Icelandic major glaciers using a comparison of MODIS snow albedo products and in situ measurements. This study could also serve as a comprehensive review of rapidly changing glaciers in Iceland with focus on impacts on their changing albedo. It brings insights into albedo analysis in problematic cloud-obscured region while providing novel findings on linear albedo gradients and dust plume shape patterns on snow and ice. Direct impacts of explosive volcanic eruptions as well as severe and moderate dust storms on the glaciers are evaluated. Additionally, possible indirect impacts of effusive eruptions such as Holuhraun 2014-2015 are suggested. It is clear that the authors know perfectly the local environment and its past. The data from the MODIS products were carefully screened during extensive manual quality control and results were evaluated with valuable data from in situ ICE-GAWS network. The greatest contribution of this study is that the data set does not only include major explosive eruptions and cold years, but it includes extremely rare year 2019, dry and dusty in the southern part of Iceland. This allows the authors to compare the impacts of volcanic ash and general volcanic/glacial dust on the albedo at the same level. There are minor errors in references that should be stated in ascending order and several references could be added. I would recommend publication after minor revisions.

Author response:

First, we would like to thank reviewer 2 (RC2) for very useful comments and a general positive feedback about our submitted manuscript. Your summary of the paper matches very well with our intended scope and deliverables.

Specific comments:

L18-95 – Introduction

Consider to add studies on snow albedo reductions due to volcanic dust, eg. **Meinander et al., 2014**, Peltoniemi et al., 2015**, Dagsson-Waldhauserova et al**., 2015, Zubko et al., 2019).

Kylling et al., 2018 calculated the instantaneous radiative forcing of the bottom of the atmosphere due to mineral dust deposited on snow as 0.135 W m-2.

Kylling A., Groot Zwaaftink, C. D., Stohl, A., 2018. Mineral dust instantaneous radiative forcing in the Arctic. Geophysical Research Letters, 45. doi: 10.1029/2018GL077346.

Peltoniemi, J. I., Gritsevich, M., Hakala, T., Dagsson-Waldhauserová, P., Arnalds, Ó., Anttila, K., Hannula, H.-R., Kivekäs, N., Lihavainen, H., Meinander, O., Svensson, J., Virkkula, A., de Leeuw, G., 2015. Soot on snow experiment: bidirectional reflectance factor measurements of contaminated snow. The Cryosphere 9, 3075-3111.

Dagsson-Waldhauserova, P., Arnalds, O., Olafsson, H., Hladil, J., Skala, R., Navratil, T., Chadimova, L., Meinander, O., 2015. Snow-dust storm A case study from Iceland, March 7th 2013. Aeolian Research 16, 69–74.

Meinander, O., Kontu, A., Virkkula, A., Arola, A., Backman, L., DagssonWaldhauserová, P., Järvinen, O., Manninen, T., Svensson, J., de Leeuw, G., and LepC2 päranta, M., 2014. Brief Communication: Light-absorbing impurities can reduce the density of melting snow. The Cryosphere 8, 991-995.

Zubko, N., Muñoz, O., Zubko, E., Gritsevich, M., Escobar-Cerezo, J., and Berg, J., 2019. Light scattering from volcanic-sand particles in deposited and aerosol form. Atmos. Env. 215, 116813. doi: 10.1016/j.atmosenv.2019.06.051

Author response:
The following references are added to introduction on impacts of dust and sand to snow covered surfaces. Meinander et al., 2014, Dagsson-Waldhauserova et al., 2015, Zubko et al., 2019, Peltoniemi et al., 2015 in L26 and onwards

L40 and L85 remove 'a' in Wittmann et al., 2017a. Consider to add Gascoin at al., 2017 here.

Author response:
We have fixed the 2017 Wittmann reference.

L40-41 – 'surface albedo IS the dominating factors' - change ARE->IS, FACTORS- >FACTOR

**Author response:**
For major revisions this paragraph was removed to account for the paper length.

In L1 in the abstract a similar sentence can be found:

*During the melt season, absorbed solar energy, modulated at the surface predominantly by albedo, is one of the main governing factors controlling surface-melt variability for glaciers in Iceland*

L44-45 – ..but it IS limited. . .

Author response:
Fixed.
L30: Optical satellite remote sensing offers a way to observe surface albedo continuously at large spatio-temporal scales but is limited to times of clear-sky overpasses.

L47 – Stroeve et al. 2001? As in reference, not 2002.

Author response:
These are two separated references, Stroeve, 2001 and Klein and Stroeve, 2002. No modifications have been made but references double checked.

(Stroeve et al., 1997; Reijmer et al., 1999; Stroeve, 2001; Klein and Stroeve, 2002; Liang et al., 2005; Stroeve et al., 2005, 2013).

L70-72 – Can you please rephrase the sentence or cut into two sentences. It is difficult to understand.

Author response:

Sentence was:
*To confirm this hypothesis, in-situ data and higher resolution data from Landsat 5 Thematic Mapper (TM) sensor were compared as well showing greater variability in surface albedo implying that the scale of the albedo variations is larger than the AVHHR pixel (1.1 km) could resolve.*

Rewrite: (L66)
*To confirm this hypothesis, Reijmer et al. (1999) compared in-situ data and higher spatial resolution remote sensing data from the Landsat 5 Thematic Mapper (TM) sensor. The results*

*showed greater variability in surface albedo, implying that the scale of the albedo variations is larger than the AVHRR pixel (1.1 km) could resolve.*

L144 – Van Den Broeke et al., 2004 a,b?

Author response:
Updated accordingly, see L156

L164 – Table 1 – What do you mean by 'average location'?

Author response:
Annually when the GAWS stations are installed in the field they are not in the exact same location from on year to another. This can vary between a few tens to hundred meters. Stations can also move during the melt season due to ice flow. We calculate the average locations, mean value of these locations for pixel data extraction instead of posting annual values.

An explanation is added in the caption text with Table 1:

*Location and elevation is based on the average location of the site for the observation period, i.e. mean location values for multi-year installations which might not be the exact same location from on year to another*

L192-193 – opening brackets are missing

Author response:
Fixed, see L 209

L229-230 – Do you mean annual melt rates here?

Author response:
…high melt rates… refers to summer melt rates indicating that large elevation changes can be expected during summer resulting in tilting of the instruments.

L230 – Sand particles have certain size resolution, maybe 'dust' is better here. Or 'sand and dust'.

Author response:
Sentence was:
*Large sand and tephra covered areas have been observed…*

Rewrite:
*Large sand, dust and tephra covered areas have been observed… see L236*

L253– Small scale spatial variability of albedo could be also discussed here. See Hartl et al., 2020. Hartl, L., Felbauer, L., Schwaizer, G., and Fischer, A.: Small scale spatial variability of bare-ice albedo at Jamtalferner, Austria, The Cryosphere Discuss., https://doi.org/10.5194/tc-2020-92, in review, 2020.

**Author response:**
The study reports a very detailed study on an small alpine glacier. The reference could fit well in an introduction or a literature review but we would question if this can be applied for Icelandic glaciers as no similar work has taken place.

L289-308 – Linear albedo gradients are really important and well discussed here. However, the role of local impurities should be also mentioned here. General lower albedo values at certain parts of Hofsjökull, Langjökull and Myrdalsjökull coincides well with location of dust source areas described in Arnalds et al., 2016, and classified as severe or extremely severe erosion areas. This should be also included here in the discussion. There is also work from Antarctica showing the vertical gradient of local dust impurities on glacier that could be discussed here. See Kavan et al., 2020.

Kavan, J., Nyvlt, D., Láska, K., Engel, Z., and Knazková, M. (2020) High latitude dust deposition in snow on the glaciers of James Ross Island, Antarctica. Earth Surf. Process. Landforms, https://doi.org/10.1002/esp.4831

**Author response:**
Correct, the following sentence is added in L 323

*Local lower albedo gradients at Hofsjökull (SE), Langjökull (S) and Mýrdalsjökull (S) coincide with documented locations of severe or extremely severe dust source areas described in Arnalds et al., 2016.*

L319-321 – General trends in annual albedo (lowest values vs. highest values) correspond to the long-term dust storm frequency studies in Iceland. For evaluation, consider these three studies:

Nakashima, M. and Dagsson-Waldhauserová, P., 2019. A 60 Year Examination of Dust Day Activity and Its Contributing Factors From Ten Icelandic Weather Stations From 1950 to 2009. Frontiers in Earth Science 6, 245-252. DOI:10.3389/feart.2018.00245

Butwin, M.K., von Löwis, S., Pfeffer, M., and Thorsteinsson, Th., 2019. The Effects of Volcanic Eruptions on the Frequency of Particulate Matter Suspension Events in Iceland. Journal of Aerosol Science 128, 99-113.

Dagsson-Waldhauserova, P., Arnalds, O., Olafsson, H., 2014. Long-term variability of dust events in Iceland. Atmospheric Chemistry and Physics 14, 13411-13422. DOI:10.5194/acp-14-13411-2014.

**Author response:**
For the cluster of glaciers (L319-321 old version) () refers to (South coast glaciers) there is no significant trend in annual values within in our study period (2000-2019), especially if the influence of the volcanic eruptions in 2010, 2011 and the residual effect in 2012 is removed.

There is a trend/pattern w.r.t. latitude generally with lower values in southern Iceland. With respect to the references provided there are discussion about longer term trends than our paper scopes.

We have added the Dagsson-Waldhauserova et al. 2014 paper in L340 and Butwin et al. 2019 in L401 to further support our findings.

L322-323 – Such unstable erosive surfaces are defined as 'dust hot spots' and it has been shown that dust events occur frequently in southern parts of Iceland in winter. Examples here:

Dagsson-Waldhauserova, P., Arnalds, O., Olafsson, H., 2014. Long-term variability of dust events in Iceland. Atmospheric Chemistry and Physics 14, 13411-13422. DOI:10.5194/acp-14-13411-2014

Dagsson-Waldhauserova, P., Renard, J.-B., Olafsson, H., Vignelles, D., Berthet, G., Verdier, N., Duverger, V., 2019. Vertical distribution of aerosols in dust storms during the Arctic winter. Scientific Reports 6, 1-11.

Dagsson-Waldhauserova, P., Arnalds, O., Olafsson, H., Hladil, J., Skala, R., Navratil, T., Chadimova, L., Meinander, O., 2015. Snow-dust storm A case study from Iceland, March 7th 2013. Aeolian Research 16, 69–74.

**Author response:**
We have added the following sentence in L340 :

*Dagsson-Waldhauserova et al., 2014, 2015, 2019 has also shown that dust events can occur frequently in southern parts of Iceland during winter given the right surface and meteorological conditions for dust transport.*

L329 – delete 'r' in severer

**Author response:**
Fixed

L329-332 – Just to comment. There are few cases when Drangajökull and Westfjords receive dust from the dust hot spots in central and South Iceland. Such events were captured by satellite or by dust model frequently in 2019.

Author response:

Figure 7 and 8 reflects this showing lower annual albedo values for 2019 for Drangajökull. In general, we would still consider Drangajökull to be "closest" of the Icelandic glaciers to have albedo driven by snow metamorphism even though dust events can take place. We have also add a sentence (See RC2 comment on L427 – Conclusions ) that highlights this.

L353 – '>30%' Did you mean < 30%

Author response:

Yes. This has been fixed, see L373

L375 – Liu et al. (2014) do not really refer to volcanic ash from eruption, but dust event with maybe some relicts of ash. Their sample was collected in 2013 and they describe a dust event in 2013. I would suggest removing this from the references here.

Author response:

Yes, agreed. We have removed the reference.

L380-381 – It was also induced by high dust storm activity in that area, see Möller et al., 2019. Volcanic ash is usually being removed fast from surfaces in Iceland, in < 1 year. See Butwin et al., 2019 or Arnalds et al., 2013.

Butwin, M.K., von Löwis, S., Pfeffer, M., and Thorsteinsson, Th., 2019. The Effects of Volcanic Eruptions on the Frequency of Particulate Matter Suspension Events in Iceland. Journal of Aerosol Science 128, 99-113.

Arnalds, O., Thorarinsdottir, E.F., Thorsson, J., Dagsson-Waldhauserova, P., Agustsdottir, A.M., 2013. An extreme wind erosion event of the fresh Eyjafjallajokull 2010 volcanic ash. Nature Scientific Reports 3, 1257.

Author response:

Sentence was:
No eruption occurred in 2012 but residual effects were observed as ash deposits from previous eruptions were carried with the prevailing wind directions, enhancing melt due to the lowering of albedo.

Rewrite: see L 399
*No eruption occurred in 2012 but residual effects were observed as ash deposits from previous eruptions were carried with the prevailing wind directions and high dust storm activity reported in the area, enhancing melt due to the lowering of albedo (Butwin er al. 2019, Möller et al., 2019).*

L382 – Wittmann et al. (2017a). Why 'a'?

**Author response:**
Duplicate in the reference list produced this, has been fixed.

Figure 8 – Correct the title – delete 'for the'?

**Author response:**
Typographical error, fixed

L400 – 1999 Hekla – Are you talking about 26th Feb 2000 Hekla eruption here?

**Author response:**
Yes, 1999 should be 2000, fixed

L386-398 – When discussing dust influence on the albedos, you can also include that not only volcanic ash can be lifted to high altitudes and transported long distances. It is also Icelandic volcanic dust that can reach several km heights and travel long distances of thousands of km:

Dagsson-Waldhauserova, P., Renard, J.-B., Olafsson, H., Vignelles, D., Berthet, G., Verdier, N., Duverger, V., 2019. Vertical distribution of aerosols in dust storms during the Arctic winter. Scientific Reports 6, 1-11.

Djordjevic D., Toši ́ c I., Sakan S., Petrovi ́ c S., Ä ́ Ruri ̌ ci ̌ c-Milankovi ́ c J., Finger D.C. and ́ Dagsson-Waldhauserová P. 2019. Can Volcanic Dust Suspended From Surface Soil and Deserts of Iceland Be Transferred to Central Balkan Similarly to African Dust (Sahara)? Frontiers in Earth Sciences 7, 142-154.

Moroni B., Ólafur Arnalds, Pavla Dagsson Waldhauserová, Crocchianti, S., Vivani R., and Cappelletti, D. 2018. Mineralogical and chemical records of Icelandic dust sources upon Ny-Ålesund (Svalbard Islands). Frontiers in Earth Science 6, 187-219.

Beckett, F., Kylling, A., SigurÃˇrardóttir, G., von Löwis, S., and Witham, C., 2017. Quantifying the mass loading of particles in an ash cloud remobilized from tephra deposits on Iceland, Atmos. Chem. Phys., 17, 4401-4418.

Ovadnevaite J., Ceburnis D., Plauskaite-Sukiene K., Modini R., Dupuy R., Rimselyte I., Ramonet R., Kvietkus K., Ristovski Z., Berresheim H., O'Dowd C.D., 2009. Volcanic sulphate and arctic dust plumes over the North Atlantic Ocean. Atmospheric Environment 43, 4968-4974

**Author response:**
This is mentioned in the manuscript, we have modified in the discussion section the following:

Sentence was.
Airborne tephra can be transported by high plumes that can extend several

kilometres into the atmosphere and be transported great distances, up to several hundred kilometers (Guðmundsson, Magnús T. et al., 2012; Watson et al., 2016)

Changed to: see L395
*Airborne tephra and dust can be transported by high plumes that can extend several kilometres into the atmosphere and be transported great distances, up to several hundred kilometers (Guðmundsson, et al., 2012; Watson et al., 2016; Đordevic et al., 2019; Dagsson-Waldhauserova et al., 2019).*

L405-417 – Can you explain better why Dyngjujökull shows positive albedo trend? Is it after the Holuhraun eruption and reduction of dust events from Dyngjusandur towards the glacier?

Author response:
Various influencing factors could contribute to a positive Dyngjujökull albedo trend. One of those possibly the changes due to the Holuhraun eruption as mentioned in L392-396 in the manuscript:

*"In 2014–15, the lava flow field of the Holuhraun non-explosive eruption covered about 84 km2 of volcaniclastic sandy desert and proglacial areas north of Vatnajökull. Since then, similar plume shaped albedo anomalies were not observed in the data. It is probable that the extent of the lava flow field reduces the dust production of this area significantly, although this cannot be quantified at this point in time, more data over a range of climatologies are needed to fully understand the impact of the Holuhraun eruption on dust production"*

But it could also be related to other climatological variables such as the variability in winter precipitation, melt onset and snowfall during late summer. The figure below shows the most recent mass balance values for Dyngjujökull among others. Since 2004-05 net mass balance has a mild upwards trend although non-significant constrained by high summer melt years for 2004 (Gjálp), 2010 (Eyjafjallajökull) and 2012 and low melt in 2015.

This is a very interesting question to discriminate the actual influencing factors driving this trend but we feel it needs a more detailed investigation than the scope of the study to be able to state anything definite.

[Figure]

Figure 16. Specific mass balance record for Vatnajökull outlets 1991_92-2018_19.

L419-426 – Figure 11 and discussion. Doesn't this show that warm, dry and dusty year as 2019 have similar impacts on albedo as volcanic eruption years?

**Author response:**
Depending on the timing of the deposits and extent this is true. In the cases where the active melting area (ablation zone) is extended significantly as happens during eruptions, more melt is expected. A key factor in the 2019 melt enhancement is how early seasonal snow is melted exposing these erosive surfaces and how early in Spring dust transport starts. Similar events are seen, although less extensive as in 2019 in 2020 but they occur mid-August, not creating a lot of melt potential for the remaining of the melt season.

L427 – Conclusions – It would be beneficial to conclude in one sentence also the difference in influence of tephra after eruption and dust during dusty year as 2019 on albedo.

Author response:
L436 says:
*Icelandic glacier albedo was observed to be influenced by variability in climate, tephra deposits from volcanic eruptions, and airborne dust from widespread unstable sandy surfaces which are subject to frequent wind erosion and dust production.*

We suggest adding the following to L447 with the discussion about Figure 11:

*Extensive dust transport to the glacier surface, as seen in 2019 melt season, had similar overall albedo lowering effect to that in the eruption years 2010 and 2011 for Vatnajökull, Langjökull, Eyjafjallajökull and Drangajökull specifically. It is, however, noted that following volcanic eruptions albedo lowering is generally more localized while extensive dust transport tends to affect larger areas.*

L472- References should be ordered in ascending order (Palsson et al., Schmidt et al., Stroeve et al, need to be corrected).

Author response:
Yes, reference list is reviewed and updated accoringly

L549-551 – Liu is not relevant reference in the text. They do not refer to volcanic ash from eruption, but general dust event. Consider to remove this from the reference list.

Author response:
The reference to his work was removed in an earlier comment

L554 – remove 'a' in Matlab, 2017a

Author response:
The "a" refers to a version number of Matlab, not multiple same year publications.

L566 – Thorsteinsson et al., 2017 should be under T in the reference list, not under P

Author response:
Corrected.

[revised manuscript text omitted]

---

## Author Response (AR2)

**Editor Decision: Publish subject to minor revisions (review by editor)** (09 Nov 2020)
by Marie Dumont
Comments to the Author:
Dear Authors,

Thanks for your effort and work on the paper.
I feel that now the paper will be ready for publication after the minor revisions below (I am copying below the referee report) :

"I apologize for the long delay in my response. I appreciate the detailed response to my comments. The authors have put some effort to revise their manuscript and I think it has improved. However, there remain a few comments listed below that were not fully addressed in my opinion.

**Author response: Publish subject to minor revisions (review by editor) (30 Nov 2020)**

Dear Marie,
Thank you for the follow-up. We fully understand delays during these strange times. Please find below our answers in green also incorporated into an updated version of the manuscript.

In addition, we have added in the Acknowledgement section:

*The authors would like to thank the editor Marie Dumont, and two reviewers, Simon Gascoin and Pavla Dagsson-Waldhauserova for their review and comments improving the manuscript.*

**1 "MCD43 provides albedo over all land masses, whereas M*D10A1 provides only albedo of the pixels that are detected as snow covered." I could not understand the response of the authors. A map illustrating one of the cases mentioned by the authors before/after the interpolation would help. In addition this issue should be much clearly acknowledge in the manuscript, I think that the added sentence is not sufficient.**

We have added the following paragraphs at line 200 to further highlight these limitations as well as added a figure in the appendix (Figure B5):

*An important aspect of M*D10A1 products is that they only provide albedo of pixels detected as snow covered while MCD43A3 provides albedo over all land masses. This does limit the application of the method presented here during or after an explosive volcanic eruption event with thick tephra depositions. Similarly, tephra plumes discharged into the atmosphere with high tephra concentrations might further induce misclassifications during explosive eruptions. In this study, during periods of volcanic eruptions, the outlier thresholds are not applied, allowing a greater range of expected albedo values, especially lower values at higher elevations where tephra deposits were observed. This applies to the melt seasons 2010 and 2011. Visual inspection of the gap-filled product during these periods was used to validate the performance of MCD11. In most cases, the method presented here was able to reconstruct albedo with acceptable accuracy, as shown in Section 3.1.*

*Figure B4 shows an example for a date after the eruption in Grímsvötn in 2011 (18.06.2020) with the original M*D10A1 products, MCD43A3 and MCD11 after median-based outlier removal and gap filling. The performance of MCD11 is good while the figure also illustrates the challenge with pixel availability for MCD4A3. The gap-filling method is capable of reconstructing albedo values in areas with thick tephra deposits.*

**2 "The improvements in MCD11 albedo with respect to the original product are very small (about 0.01 RMSE, Tab.2)." Since the authors agree I believe they should also make it clear it in the manuscript and emphasize that the main benefit of this work is the gap filling.**

We do not assert in the manuscript that MCD11 is generally better or worse than MCD43 or M*D10A1. The objectives of our study are detailed in lines 100-107, as well as why M*D10A1 is selected rather than MCD43. This was further addressed in the major review of the manuscript. L95-98 also mentions the scope and intention of the work. We do not expect that the performance of our results is better than that of either M*D10A1 or MCD43A3 where data from all products are available. Tables 2 and B1 show a detailed comparison. At some locations and time periods improvements are observed while in others not. We think the reason behind the work is explained in the introduction section.

To respond to the comment we add in the conclusion (Line 489):

*The good visual and statistical agreement that was found between the MCD11 data and in-situ albedo from GAWS observations indicates that the gap-filling method applied in this study is able to provide good quality daily albedo estimates both spatially and temporally. This illustrates the main strength of the spatio-temporal MCD11 data set, which is obtained without sacrificing quality compared with other data sources.*

**3 ""Daily averages" is not the correct wording". From the first version of the manuscript I understood that the daily albedo was computed as the ratio of the daily sums of incoming and reflected radiation. From the author response ("Daily averages were calculated from hourly averages if at least 20 hourly values were available") I understand that the daily albedo was computed from the average of the hourly albedos. This should be clarified because it is not the same. The second method gives equal weight to albedo measurements in the morning or in the evening. The first method is less affected by low sun elevation periods.**

Daily albedo is computed as the ratio of the daily sums of incoming and reflected radiation. This is stated in L148-150:

*Daily integrated albedo was calculated as the running 24-hour sum of upward shortwave divided by the running 24-hour sum of the downward shortwave.*

I think what is unclear here and poorly worded by us, is that albedo is not calculated, as daily sums, unless there are enough values to base the calculations on. We might have misunderstood the original comment. We suggest adding the original sentence, removed during major reviews, changing *Daily averages* to *Daily values* (Line 167):

**Daily albedo values were calculated from sums of hourly radiation averages if within each day at least 20 hourly values or more were available and monthly averages were calculated from daily values if within each month 24 daily values or more were available.**

**4 "median based statistical rejection of outliers.": the outlier rejection paragraph remains a bit obscure to me. I understand that values outside the interval [median-STD ; median-STD] are rejected, except if the STD is lower than x% of the median (where x ranges from 1% to 4%).**

We have added the following in line 195 to clear this up (the range should be [2*median-STD ; median-2*STD] representing approximate 95% confidence interval):

*This means that data from the 11-day temporal aggregate were not rejected, even if the difference between the albedo value and the median was greater than 2 standard deviations, if the difference falls within the pixel-defined threshold value.*

**5 "results are not directly comparable (daily vs. monthly)". It would have been interesting to provide the results at the daily time step too (which is much more informative and discriminant in terms of RMSE) but I leave it up to the authors to consider if they have the time!**

At this point in time we are preparing a manuscript using MCD11 on daily timesteps to reconstruct seasonal snow and glacier melt for Iceland. It uses daily albedo data. In light of this, we chose not to incorporate analysis of daily values in this manuscript.

**6 "Fig 3: a similar figure can be found in Gascoin et al 2017". If the authors keep the figure then they should explain in their paper (not only in the response the reviewer) what is here that was not there."**

We have added a paragraph in Line 296:

[revised manuscript text omitted]